# Linking post-translational modifications and protein turnover by site-resolved protein turnover profiling

Jana Zecha [1,2,6], Wassim Gabriel [1,7], Ria Spallek [3,4], Yun-Chien Chang[1], Julia Mergner [1,8], Mathias Wilhelm [1,7], Florian Bassermann[2,3,4] & Bernhard Kuster [1,2,5 ✉]

Proteome-wide measurements of protein turnover have largely ignored the impact of post-translational modifications (PTMs). To address this gap, we employ stable isotope labeling and mass spectrometry to measure the turnover of >120,000 peptidoforms including >33,000 phosphorylated, acetylated, and ubiquitinated peptides for >9,000 native proteins. This site-resolved protein turnover (SPOT) profiling discloses global and site-specific differences in turnover associated with the presence or absence of PTMs. While causal relationships may not always be immediately apparent, we speculate that PTMs with diverging turnover may distinguish states of differential protein stability, structure, localization, enzymatic activity, or protein-protein interactions. We show examples of how the turnover data may give insights into unknown functions of PTMs and provide a freely accessible online tool that allows interrogation and visualisation of all turnover data. The SPOT methodology is applicable to many cell types and modifications, offering the potential to prioritize PTMs for future functional investigations.

[1] Chair of Proteomics and Bioanalytics, Technical University of Munich (TUM), Freising, Germany. [2] German Cancer Consortium (DKTK) and German Cancer Research Center (DKFZ), Heidelberg, Germany. [3] Department of Medicine III, Klinikum rechts der Isar, TUM, Munich, Germany. [4] TranslaTUM, Center for Translational Cancer Research, TUM, Munich, Germany. [5] Bavarian Biomolecular Mass Spectrometry Center (BayBioMS), TUM, Freising, Germany. [6] Present address: Dynamic Omics, Centre for Genomics Research, Discovery Sciences, R&D AstraZeneca, Gaithersburg, MD, USA. [7] Present address: Computational Mass Spectrometry, Technical University of Munich (TUM), Freising, Germany. [8] Present address: Bavarian Biomolecular Mass Spectrometry Center at Klinikum rechts der Isar (BayBioMS@MRI), TUM, Munich, Germany. ✉email: kuster@tum.de

Ever since the continuous turnover of body constituents was discovered eight decades ago[1], the dynamic nature of cellular proteins has been studied extensively[2], and its significance to maintain protein homeostasis has been widely acknowledged[3–5]. More recently, so-called dynamic or pulsed SILAC (stable isotope labeling by amino acids in cell culture) approaches together with mass spectrometry-based readouts have enabled the investigation of endogenous protein turnover on a proteome-wide scale[6]. Turnover of cellular proteins is determined by tracing the incorporation and loss of isotopically labeled amino acids into proteins in a time-resolved manner. Protein half-lives computed from such data have been found to range from minutes to weeks and vary considerably between cell types and tissues[7–9]. This demonstrates the high degree of regulation exerted on protein synthesis and degradation processes. For some proteins, post-translational modifications (PTMs) have been implicated in regulating protein stability. Most prominently, polyubiquitin chains linked via the side chain of lysine at position 48 typically mark proteins for proteasomal degradation[10]. Phosphorylated motifs that serve as recognition elements for certain E3 ubiquitin ligases can mediate ubiquitination and subsequent protein degradation. Such phosphodegrons are, for instance, responsible for the periodic degradation of cell cycle regulators to enable the progression of cell proliferation[11,12]. In contrast to such positive PTM crosstalk, a competitive interplay between stabilizing acetylation and destabilizing ubiquitination has been suggested for several transcription factors including p53 and MYC[13–15]. Yet, acetylation alone can also result in the degradation of other transcriptional regulators, such as HIF1α and GATA-1[16,17] which illustrates the diversity and complexity of the underlying regulatory mechanisms. Importantly, for most proteins, the types of modifications that may govern protein degradation and the sites on which they occur, remain elusive demonstrating the need for a thorough global interrogation of links between PTMs and protein turnover.

To begin to address this, we employ two workflows that we abbreviate as site-resolved protein turnover (SPOT) profiling to create a large dataset of endogenous protein turnover with PTM resolution. Phosphorylation, acetylation, and ubiquitination SPOT profiling suggests that modification sites that differ in turnover from the underlying bulk protein are involved in diverse cellular processes that go beyond immediate protein stabilization or destabilization. To illustrate the utility of our data as a resource and to highlight PTMs of potential regulatory relevance, we discuss examples of sites that may be involved in protein interactions, localization, enzymatic activity or degradation.

## Results

### SPOT analyses detect PTM-associated differences in protein turnover.

Two different dynamic SILAC (dSILAC) methods were employed to investigate associations between PTMs and protein turnover. For both, we determined the turnover of (un)modified peptides in a bottom-up proteomics approach and utilized it as a proxy for the turnover of (un)modified proteins. To this end, HeLa cells were pulsed with SILAC medium in quadruplicates including two label swap experiments: For two replicates, medium containing light lysine (K0) and arginine (R0) on unlabeled cells was exchanged with medium containing $^{13}C$ and $^{15}N$ labeled, heavy lysine (K10) and arginine (R10; exemplified in Fig. 1a, b). For the other two replicates, K8R10 medium on fully K8R10 labeled cells was replaced with K0R0 medium.

In the first SPOT approach (dynamic SILAC/dSILAC), we lysed cells at three time points (6, 24, 40 h). Proteins were digested, and phosphorylated, di-glycine (ubiquitin-remnant)-modified and acetylated peptides were sequentially enriched

followed by the quantification of light and heavy labeled peptides by mass spectrometry (Fig. 1a). The flow-through of the sequential enrichment was used for whole proteome analysis. Subsequently, we utilized SILAC intensity ratios—representing the ratio of newly synthesized to pre-existing peptide species (N/P ratio)—to compare their turnover within a particular pulse period (Supplementary Data 1). Shorter SILAC pulses were investigated but not included in the data analysis because they exhibited substantial amino acid recycling and inferior correlations between SILAC label swap experiments implicating erroneous turnover estimations (Supplementary Fig. 1).

In the second SPOT approach (dSILAC-TMT), we multiplexed 10 pulse time points (0, 1, 3, 6, 10, 16, 24, 34, 48, >200 h) using TMT (tandem-mass tags) and performed sequential enrichments of phosphorylated and acetylated peptides to measure their time-resolved SILAC labeling behavior and to obtain absolute turnover rates (K, Fig. 1b, Supplementary Data 1). Again, the flow-through of the sequential enrichment was used for whole proteome analysis. Turnover data quality and depth were considerably improved compared to our previously published dynamic SILAC-TMT strategy[7] by computational removal of TMT ratio compression, adjustment for changes in peptide abundance during the pulse time-course, and refinement of filter criteria for fitted turnover curves (Supplementary Fig. 2, Online Methods). For a comparative discussion on the two stable isotope labeling workflows, see Supplementary Discussion.

In the following, we use the term 'peptidoforms' for the entirety of peptides with and without post-translational modifications (without counting different oxidized versions of peptides repeatedly). Further, we refer to 'counterpart peptides' to describe peptidoforms that contain an unmodified amino acid that was identified with a PTM in another peptidoform. In total, turnover characteristics were determined for 120,390 peptidoforms representing 9,073 human genes (9,154 UniProt entries, 19,531 isoforms; Fig. 1c). PTM-specific turnover was obtained for 33,884 modified peptidoforms covering 29,237 modification sites (Fig. 1d, Supplementary Fig. 3). All data of modified and counterpart sites can be viewed and explored via the SPOT web tool (spot.proteomics.wzw.tum.de).

Distributions of N/P ratios and turnover rates (Fig. 2a, Supplementary Fig. 4) revealed an overall slightly faster turnover of phospho-peptidoforms compared to unmodified peptides. This shift, however, vanished when the phospho-peptidoforms' turnover was normalized to the turnover of the corresponding protein. This indicates that phospho-proteoforms do not commonly involve an increased protein turnover, but are more frequently identified on higher turnover proteins. As expected based on the well-established involvement of ubiquitination in proteasomal protein degradation, di-glycine modified peptidoforms also showed a globally increased turnover. This was particularly noticeable at early pulse time points and even increased after normalization to the turnover of their proteins or counterpart peptides. We note that human NEDD8 and ISG15 would leave the same diGly signature. However, it has been estimated that ~95 % of the diGly-remnant sites originate from ubiquitination[18]. Distributions of acetylated peptidoforms were most distinct from all others in that a large fraction featured a considerably slower turnover compared to other peptidoforms and their corresponding proteins.

For each of the four datasets, we assessed the statistical significance of turnover differences between (modified) peptidoforms and their corresponding proteins or counterpart peptides (Supplementary Fig. 5, Supplementary Data 2, Online Methods). Collectively, 10,050 peptidoforms assigned to 2,788 UniProt entries (6,279 isoforms) scored in at least one of the statistical tests (Supplementary Data 3). This corresponds to 11.5% of all

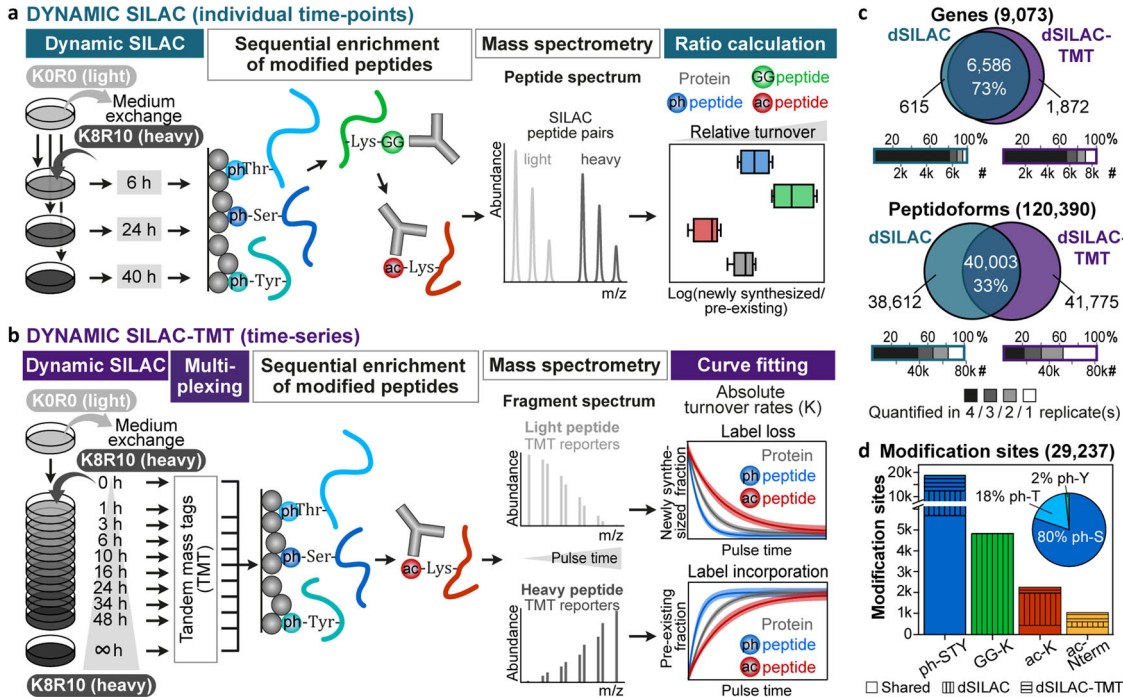

**Fig. 1 Site-resolved protein turnover (SPOT) analysis. a** Dynamic SILAC (dSILAC) workflow. Unlabeled HeLa cells (K0R0: light lysine and arginine) are lysed after a pulse with K8R10 (heavy lysine and arginine) containing SILAC (stable isotope labeling by amino acids in cell culture) medium, followed by protein digestion and enrichment of modified peptides. Peptides originating from newly synthesized proteins after the pulse start (heavy peptides) can be discriminated from peptides derived from pre-existing proteins (light peptides) via their mass shift and are quantified in the mass spectrometer. SILAC ratios (heavy/light = newly synthesized/pre-existing) are utilized as proxy for the turnover of proteins and peptidoforms (modified and unmodified peptides). **b** Dynamic SILAC-TMT workflow. Peptides derived from different SILAC pulse time points are multiplexed using TMT (tandem-mass-tags) before enrichment of post-translational modifications. Absolute turnover rates are obtained from curve fitting to time-resolved behavior of label loss (decrease of pre-existing proteins) and incorporation (increase of newly synthesized proteins), which are determined via mass spectrometry. **c** Overlap of genes and peptidoforms and the number of cell culture replicates for which relative turnover (dSILAC) and absolute turnover rates (dSILAC-TMT) were determined. Oxidized versions of peptides were not counted separately. **d** Overlap of different modification types for which relative turnover (dSILAC) and absolute turnover rates (dSILAC-TMT) were obtained (ph-S/T/Y: phosphorylated serine/threonine/tyrosine; GG-K: lysine with ubiquitin-remnant; ac-K: acetylated lysine; ac-Nterm: acetylated protein amino terminus). The pie chart shows the fractions of phospho-serine, -threonine, and -tyrosine residues.

peptidoforms tested. Less than 0.1 % showed a significant regulation in opposing directions in different datasets attesting to the robustness of the data. Significant peptidoforms comprised 4,036 modified sites (20.6 % of all tested) with less than 0.8 % exhibiting an opposing regulation on different peptidoforms or in different datasets (Supplementary Data 3). In line with $N/P$ and turnover rate distributions, the fraction of PTMs with significantly slower or faster turnover varied considerably between modification types (Fig. 2b).

**Differential site turnover highlights PTMs with diverse functions**. While many PTMs can be efficiently detected by mass spectrometry, most do not yet have an ascribed function. Within our dataset, we found that about 20 % of all annotated regulatory sites (according to PhosphoSitePlus), exhibited a differential turnover behavior (Supplementary Data 3). However, most PTMs on peptidoforms for which we observed differential turnover currently lack functional annotations (Fig. 2b). Among those with a known regulatory function, we identified sites that have been reported to change cellular protein half-lives (e.g. LATS1_ph-S464[19], GRB10_ph-S428[20], Fig. 2c, d). Strikingly, PTMs that have been described to regulate enzymatic activity (e.g. PTGES3_ph-113[21], RPS3_ph-T221[22]), protein localization (e.g. GAPDH_ac-K227/251[23], RPS3_ph-T221[22]), or interactions with other proteins (e.g. PTGES3_ph-113[21], PDHA1_ac-K321[24]) also showed significant turnover differences (Fig. 2e–h). While one may not necessarily be able to generalize from examples, this indicates that

SPOT analyses can detect modification sites with diverse regulatory functions beyond degradation or stabilization signals. Of note, many PTMs lacking functional annotations featured even larger turnover differences than sites with known regulatory functions (e.g., GAPDH_ac-K84/139/215, RPS3_GG-K132/227, Fig. 2f, g) emphasizing the potential of SPOT profiling for the discovery of functionally important modification sites.

**PTMs with differential turnover are enriched in distinct proteasomal fractions**. We next asked whether peptidoforms with differential turnover are overrepresented in certain domains, protein classes, pathways, or complexes and found an overrepresentation of ontologies pointing to cytoskeletal components, metabolic proteins, chaperone pathways, and large protein complexes such as the ribosome, spliceosome, and proteasome (Supplementary Data 4). For example, 20 % of the tested peptidoforms (57 % of modified versions) that map to the 26 S proteasome scored in at least one of the statistical tests. The corresponding modification sites were mainly found on the α-ring of the 20 S core and the base of the 19 S regulatory particle (Supplementary Fig. 6a, b). Surprisingly, the PhosphoSitePlus database did not provide regulatory functions for any of the identified PTMs. However, based on previous reports that the turnover of complex subunits can differ between the monomer and complex state[25,26] and our observation that many sites are located at interaction surfaces (PDB 5LE5[27], Supplementary Fig. 6c), we hypothesize that these PTMs may represent different proteasome assembly states. We further noticed some overlap with PTM sites that are up-

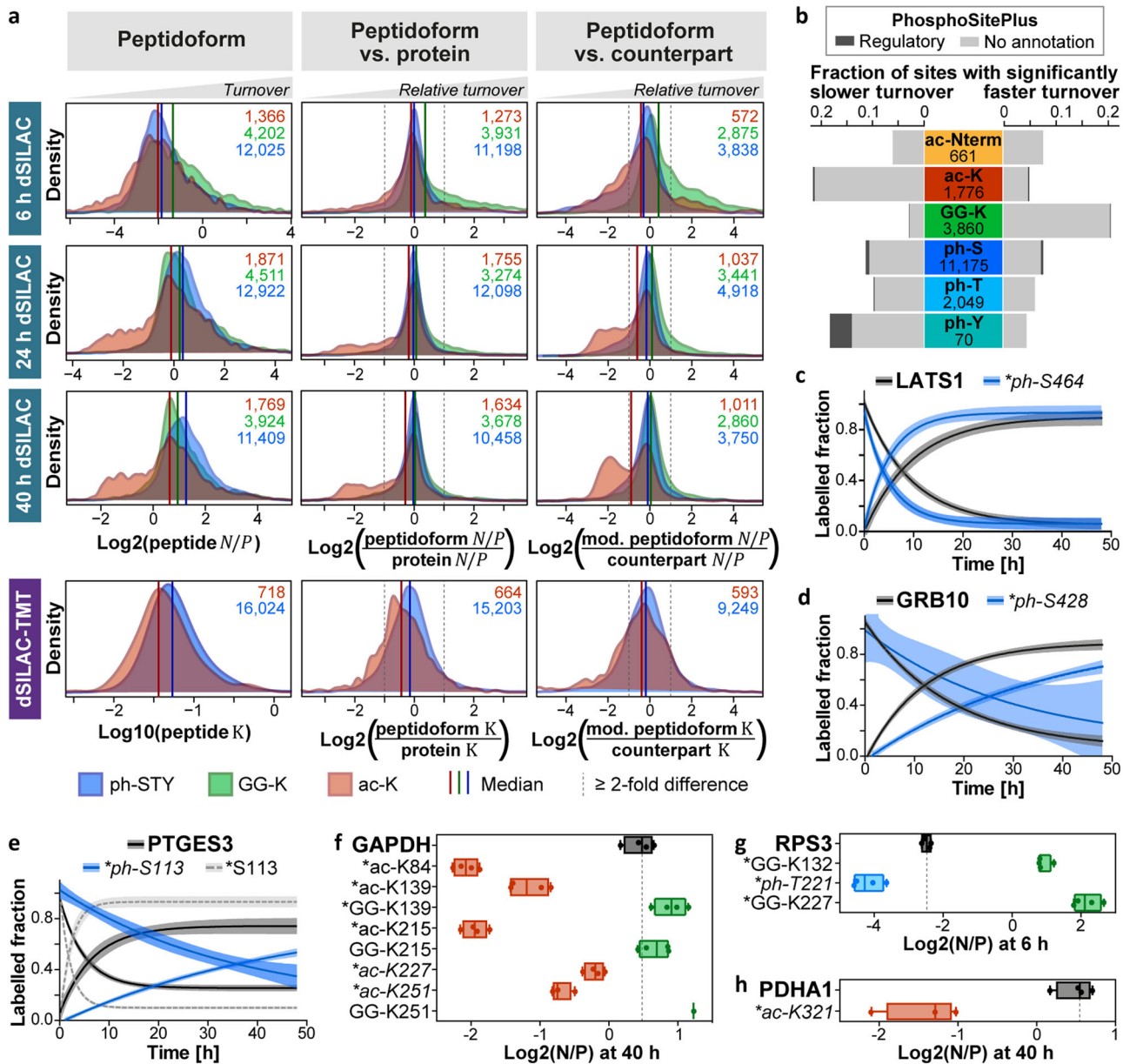

**Fig. 2 Global and site-specific associations of PTMs and protein turnover. a** Distributions of newly synthesized-to-pre-existing (N/P) ratios and turnover rates (K) for modified peptidoforms (left column) and distributions of their N/P ratios and K values relative to the N/P ratios and K values of their corresponding proteins (middle column) or unmodified counterpart peptides (right column). Ratios to proteins and counterparts were calculated based on the median of log2 values across $n = 4$ cell culture replicates. For the comparison to the protein turnover, only proteins with at least three peptides were included. Numbers of included peptidoforms are given in the top right corner (blue/ph-STY: phosphorylated serine/threonine/tyrosine; green/GG-K: lysine with ubiquitin-remnant; red/ac-K: acetylated lysine). **b** Fraction of modified sites showing a significantly slower or faster measured turnover compared to the corresponding protein or unmodified counterpart site. Total numbers of statistically tested PTMs are indicated. The fraction of sites with functional annotations according to the PhosphoSitePlus database are shown in dark gray. **c-e** Turnover curves for three example proteins and their phosphosites. Displayed sites have been described to increase protein degradation (LATS1_ph-S464), stabilize the protein (GRB10_ph-S428), or enhance enzymatic activity and protein interaction with Hsp90 (PTGES3_ph-S113). Shaded areas illustrate 95 % confidence intervals and lines represent means of $n = 4$ cell culture replicates, except for LATS1 label loss, GRB10_ph-S428 label incorporation (both $n = 3$), LATS1_ph-S464 label loss ($n = 2$), and GRB10_ph-S428 and PTGES3_ph-S113 label loss (both $n = 1$). **f-h** Tukey-boxplots of log2 N/P ratios for three example proteins and selected modified sites (box borders: 1st and 3rd quartile; lines in boxes: medians; whiskers: ranging to greatest value within 1.5× interquartile range; dots: replicate measurements; dashed line: median of protein N/P; *: significantly different turnover to protein or unmodified counterpart). Known regulatory sites are italicized and have been reported to cause nuclear translocation (GAPDH_ac-K227/251, RPS3_ph-T221), induce endonuclease activity (RPS3_ph-T221), or regulate the interaction with PDK1 (PDHA1_ac-K321). Boxplots are based on $n = 4$ cell culture replicates, except for GAPDH_ac-K251, PDHA1_ac-K321 (both $n = 3$), and GAPDH_GG-K251 ($n = 1$). PTM positions are given for the major protein isoform. For panels **c–h**, the number of peptides and spectra included varies for each site/protein and replicate and can be retrieved from Supplementary Data 1. Source data are provided as a Source Data file.

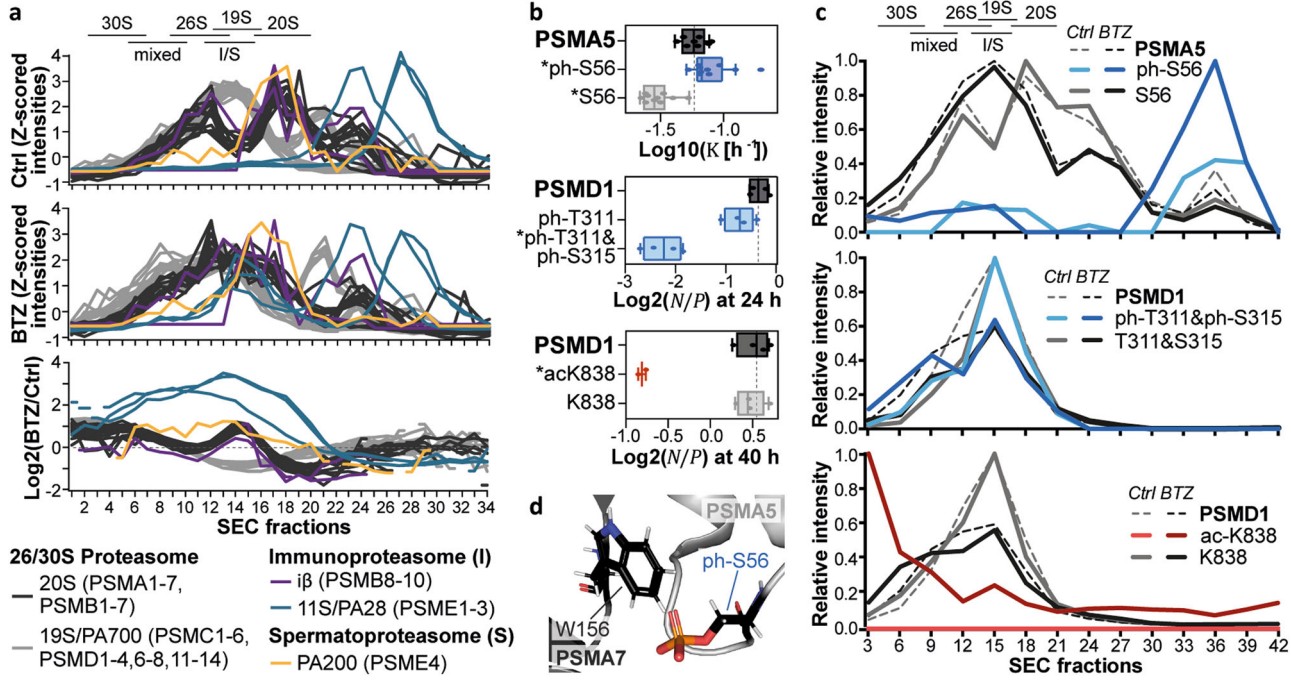

**Fig. 3 PTMs in different complex and monomer fractions of the proteasome. a** Size exclusion chromatography (SEC) profiles of proteasomal subunits in RPMI8226 cells upon proteasome inhibition ($n = 1$ biological replicate, see Supplementary Fig. 7 for an additional HeLa cell replicate). The upper and middle panel show profiles from control (Ctrl) and Bortezomib (BTZ)-treated cells (1 μM, 16 h), respectively. The lower panel displays changes upon proteasome inhibition as log2 ratios. Fractions containing proteasome complexes are indicated based on extrapolation of lower molecular weight markers (mixed: 20 S + 19 S + 11 S or 20 S + 19 S + PA200). **b** Tukey-boxplots of log10 K values or log2 N/P ratios for selected proteasomal subunits and modification sites (box borders: 1st and 3rd quartile; lines in boxes: medians; whiskers: ranging to greatest value within 1.5× interquartile range; dots: replicate measurements; dashed line: median of protein N/P; *: significantly different turnover to protein or unmodified counterpart). Boxplots are based on $n = 4$ cell culture replicates, except for PSMD1_ac-K838 ($n = 2$). For PSMA5 boxplots, the number of peptides and spectra varies for each replicate and can be retrieved from Supplementary Data 1. **c** SEC profiles for proteins, modified and unmodified sites displayed in panel b with and without proteasome inhibition ($n = 1$ biological replicate, see Supplementary Fig. 7 for an additional HeLa cell replicate). Peptide profiles were acquired via targeted parallel-reaction monitoring assays using pooled SEC fractions, while protein intensities from whole proteome measurements were combined in silico for every three adjacent fractions. **d** Clash of phosphorylated PSMA5 S56 with PSMA7 W156 within the α-ring of the proteasome (PDB 5LE5[27]). The PyMol plugin PyTMs was used to add the phospho-group to the serine residue. Source data are provided as a Source Data file.

regulated upon proteasome inhibition from previous experiments in our laboratory. Hence, we treated HeLa and RPMI8226 cells with proteasome inhibitors and subjected the lysates to size exclusion chromatography (SEC). Whole proteome measurements of the SEC fractions indeed revealed a change in the abundance of different proteasome complexes after proteasome inhibition with a general shift towards higher molecular weight complexes (Fig. 3a, Supplementary Fig. 7a, Supplementary Data 5). Solitary 20 S core particles decreased in abundance and appeared to assemble with PA28 and PA200, the regulatory subunits of the immune- and spermatoproteasome. Likewise, proteasomes containing two rather than one regulatory subunit increased in abundance.

Additional measurements using parallel-reaction monitoring (PRM) allowed us to robustly quantify three modified peptidoforms of PSMA5 and PSMD1 and their counterparts. Phosphorylation of S56 was associated with higher PSMA5 turnover and was predominantly detected in SEC fractions where the monomer or assembly intermediates would be expected (Fig. 3b, c, Supplementary Fig. 7b, Supplementary Data 5). Published crystal structures suggest that the van-der-Waals radii of PSMA5 ph-S56 and PSMA7 W156 overlap (~0.8 Å distance, Fig. 3d). This may imply some structural rearrangement to permit α-ring assembly or may give a hint that its assembly may be impaired by phosphorylation on PSMA5 S56. In contrast, ac-K838 and ph-T311/S315 were linked to reduced PSMD1 turnover. The doubly phosphorylated peptide appeared only slightly enriched in larger proteasomes compared to

its counterpart peptide, but the acetylated peptide was strongly overrepresented in higher molecular weight complexes. All three modifications reside in regions with high structural flexibility that are not covered by the PDB x-ray structures of the proteasome. The underlying molecular mechanisms of the PTM enrichment in distinct monomer and complex fractions remain to be resolved in more detail. Yet, even if those PTMs may not be causal for the formation or disassembly of the proteasome, these examples illustrate that modified peptidoforms with differential turnover can be markers for different assembly states of protein complex subunits.

**Acetylation and ubiquitination show opposing associations with turnover**. It has been hypothesized that lysine acetylation stabilizes certain proteins by preventing ubiquitination on the same residue[13,14]. While this is an intriguing concept, it is unclear to what extent this may be generalized. Enrichment analyses using our dSILAC data indicated that 411 acetylation-ubiquitination pairs were rarely located in membrane proteins but overrepresented in glycolytic enzymes, the ribosome, chaperones, the cytoskeleton, and RNA-binding proteins (Fig. 4a, Supplementary Data 4). To quantify the impact of either modification, we compared the turnover of acetylation and ubiquitination sites to their unmodified counterparts. Since $N/P$ ratios are specific for each pulse duration, ratios to counterparts were only calculated for the same pulse time. For the resulting 252 lysine residues on 150 proteins, acetylation was more often

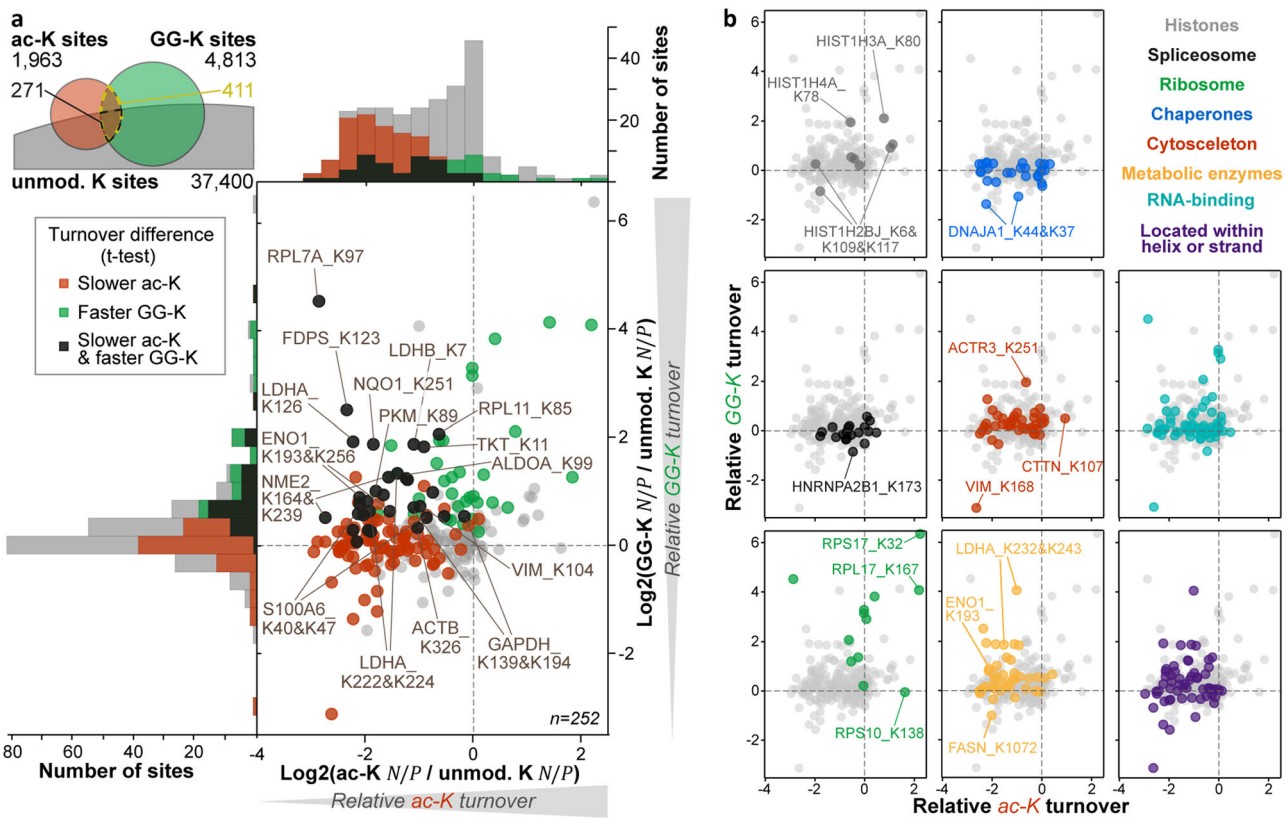

**Fig. 4 Turnover interplay of acetylation and ubiquitin-remnant sites. a** Scatter plot and distributions of the turnover of acetylated (ac-K) and ubiquitin-remnant (GG-K) peptidoforms relative to their counterpart as measured by dynamic SILAC. The average of all modified peptidoform-counterpart comparisons in all three pulse time points is plotted. Sites with significant turnover differences in at least one t-test are colored in red, green, and black. The number of pulse time points, cell culture replicates and peptides included for each displayed dot varies and can be retrieved from Supplementary Data 1 and 6. The Venn diagram indicates how many acetylated, ubiquitinated, and unmodified lysine residues share the same position in the dynamic SILAC datasets. **b** Same scatterplots as in panel a but colored according to their affiliation to various protein categories or their location within ordered secondary protein structures. Source data are provided as a Source Data file.

connected to a significantly reduced protein turnover than ubiquitination was associated with an increased turnover (Fig. 4a, Supplementary Data 6). Sites that showed both effects were frequently located on metabolic enzymes. Interestingly, the acetylation-ubiquitination interplay generally appeared to differ between protein classes (Fig. 4b). Similar to metabolic enzymes, acetylated cytoskeletal and RNA-binding proteins commonly exhibited a slower turnover, and ubiquitinated versions featured a faster turnover. While acetylation was also associated with decreased turnover of many spliceosome subunits and chaperones, their ubiquitination rarely had any impact. In contrast, ubiquitination of most ribosomal subunits was strongly linked to enhanced turnover. Acetylated RPL7A (ac-K97), however, showed a slower turnover, but acetylated RPS10 (ac-K138), RPS17 (ac-K32), and RPL17 (ac-K167) exhibited an increased turnover. For histones, the turnover of acetylated and ubiquitinated versions changed in both directions. Taken together, the above illustrates that the interplay of acetylation and ubiquitination on identical lysine residues is protein-specific. Moreover, the more frequent shift in turnover of acetylated peptidoforms suggests that the presumed stabilizing effect of acetylation may not exclusively be explained by an inhibition of apparently destabilizing ubiquitination.

**Differential acetylation turnover may describe distinct protein states.** To further characterize modification sites associated with slower and faster turnover, we conducted sequence motif

analyses[28,29] (Fig. 5a, Supplementary Fig. 8a, b). For ubiquitination sites, no clear motif could be extracted. In contrast, for acetylation and phosphorylation sites with a reduced turnover, proline was significantly enriched in the +1 position. This may suggest that phosphorylation by proline-directed kinases may stabilize their substrates[30] or otherwise slow down their measured turnover. For acetylation sites, we could not identify any similarity to a probability logo generated from substrates of the lysine acetyltransferase (KAT) CBP/p300[31] (Supplementary Fig. 8c). Strikingly, however, we found that the combined sequence logo of sites regulated by 19 lysine deacetylase (KDAC) inhibitors[32] mirrored our logo of ac-K sites that were linked to slower turnover. In other words, amino acids that were overrepresented in one motif were under-represented in the other (Fig. 5a, b). Thus, one may imagine a mechanism by which acetylation on lysine residues followed by a proline can accumulate on aged proteins because the KDACs appear less active on these acetyl moieties (Fig. 5c). Such a disequilibrium of enzymatic activities of the writer and eraser enzymes can be detected by SPOT analysis as it would manifest in slower turnover.

Despite a strong overrepresentation of ac-KP sites among ac-K sites with decreased turnover, 87 % of the latter were not followed by proline. The glycolytic enzymes GAPDH, ENO1, and ALDOA, for instance, contained the most ac-K sites with an apparent slower turnover (10, 9, and 8, respectively). None of these are functionally annotated in PhosphoSitePlus, and only two feature a proline in +1 position (Supplementary Data 3). Remarkably, however, two-thirds

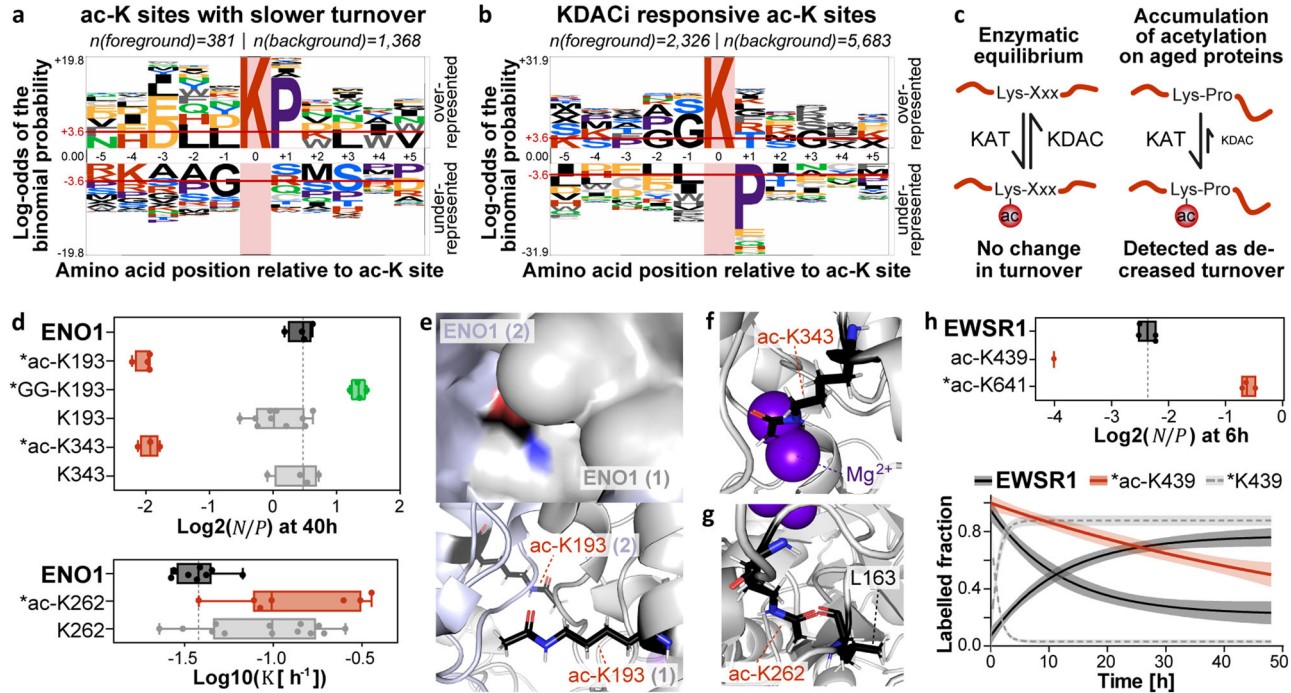

**Fig. 5 Associations of differential ac-K turnover with structural and enzymatic events. a–b** Probability logos of acetylated lysine (ac-K) sites linked to slower turnover and ac-K sites responsive to lysine deacetylase inhibitors (KDACi) reported by Schölz et al. (red horizontal line: Bonferroni-corrected p-value of 0.05 indicating significant over- or underrepresentation). For the KDACi motif, only sites that showed at least a TWOfold regulation upon KDAC inhibition were included. Logos were generated using pLogo[29] (https://plogo.uconn.edu/). The background was defined by all sites quantified in the respective studies. **c** Potential model explaining the measured decrease in turnover on ac-K sites followed by a proline (KAT: lysine acetyltransferase; KDAC: lysine deacetylase). **d** Tukey-boxplots of log2 N/P ratios or log10 K values for ENO1 and selected lysine sites (box borders: 1st and 3rd quartile; lines in boxes: medians; whiskers: ranging to greatest value within 1.5× interquartile range; dots: replicates; dashed line: median of protein N/P; *: significantly different turnover to protein or unmodified counterpart). Boxplots are based on n = 4 cell culture replicates, except for ac-K343 (n = 2). **e–g** Positioning of ac-K139, ac-K262, and ac-K343 in the ENO1 homodimer (PDB 3B97[33]; black: carbon; red: oxygen; blue: nitrogen; gray: hydrogen; purple: Mg2+ ions). The PyMol plugin PyTMs was used to add acetyl moieties to lysine residues. **h** Tukey-boxplots of log2 N/P ratios (description same as in panel d) and turnover curves (solid lines: means; shaded areas: 95 % confidence intervals) for EWSR1 and selected lysine sites. Plots are based on n = 4 cell culture replicates, except for ac-K641 (n = 3) and the box and label loss curve of ac-K439 (n = 1 and n = 2, respectively). K439 and K641 are located in the RNA-recognitions motif and the nuclear localization signal, respectively. For panels d and h, the number of peptides and spectra included in each plot varies for each site/protein and replicate and can be retrieved from Supplementary Data 1. Positions of post-translational modifications are given for the major protein isoform. Source data are provided as a Source Data file.

of those are located within α-helices or β-sheets, sequence elements that were generally enriched among ac-K sites involving slower turnover (Fig. 4b, Supplementary Data 4). We picked ENO1, the protein with the most differentially turned over ac-K sites, to examine potential structural effects of lysine acetylation in silico (PDB 3B97[33]). For instance, K193 is located at the interface of the homodimer, and acetylation may facilitate the interaction between both molecules providing a possible hypothesis for its presumed stabilizing effect (Fig. 5d, e). Ubiquitination of this residue, in contrast, may prevent ENO1 dimerization potentially leading to an increased turnover. We also detected a slower turnover of ac-K343. This residue functions as a proton acceptor in the conversion of 2-phosphoglycerate to phosphoenolpyruvate[34]. Acetylation would be expected to reduce the proton acceptor capability of this site and may, as a result, displace one of the Mg$^{2+}$ ions that are required for catalysis (Fig. 5f). Therefore, acetylation of K343 possibly impairs the enzymatic activity of ENO1. Site-directed mutagenesis experiments would be required to test if this is indeed the case. The protein structure analysis further suggested that ac-K262, which exhibited a slightly accelerated turnover, clashes with the carbonyl oxygen of L163 (1.1 Å between the oxygen atoms, Fig. 5g). Moreover, acetylation would break the hydrogen bond that presumably forms between the ε-amino group of

K262 and the carbonyl group of L163 in the non-acetylated state (2.8 Å between the nitrogen and oxygen atom). Together, this may elicit some change within the ENO1 structure to allow acetylation to take place on K262. These examples indicate that a variety of enzymatic and structural events associated with acetylation may alter a protein's turnover as measured by SPOT profiling.

While acetylation was globally linked to reduced turnover, we identified certain acetylation events that showed the opposite effect. One example is ac-K641 in the nuclear localization signal (NLS) of EWSR1 (Fig. 5h). Interestingly, mutations in amino acids leading to the removal of positive charges in the NLS of this proto-oncogene have been shown to prevent nuclear translocation[35]. Therefore, one may speculate that acetylation of K641 may also alter EWSR1 localization. Another ac-K site located in the RNA-recognition motif at position 439 showed a considerable decrease in EWSR1 turnover (Fig. 5h). This may hint towards a potential regulation of RNA binding via lysine acetylation. More importantly, the large difference in turnover measured for the ac-K439 and ac-K641 peptidoforms demonstrates that these two acetylation events cannot co-occur on the same protein molecule. Hence, SPOT profiling can provide information about the coexistence of very distant PTM sites, which typically cannot be obtained from bottom-up proteomics data.

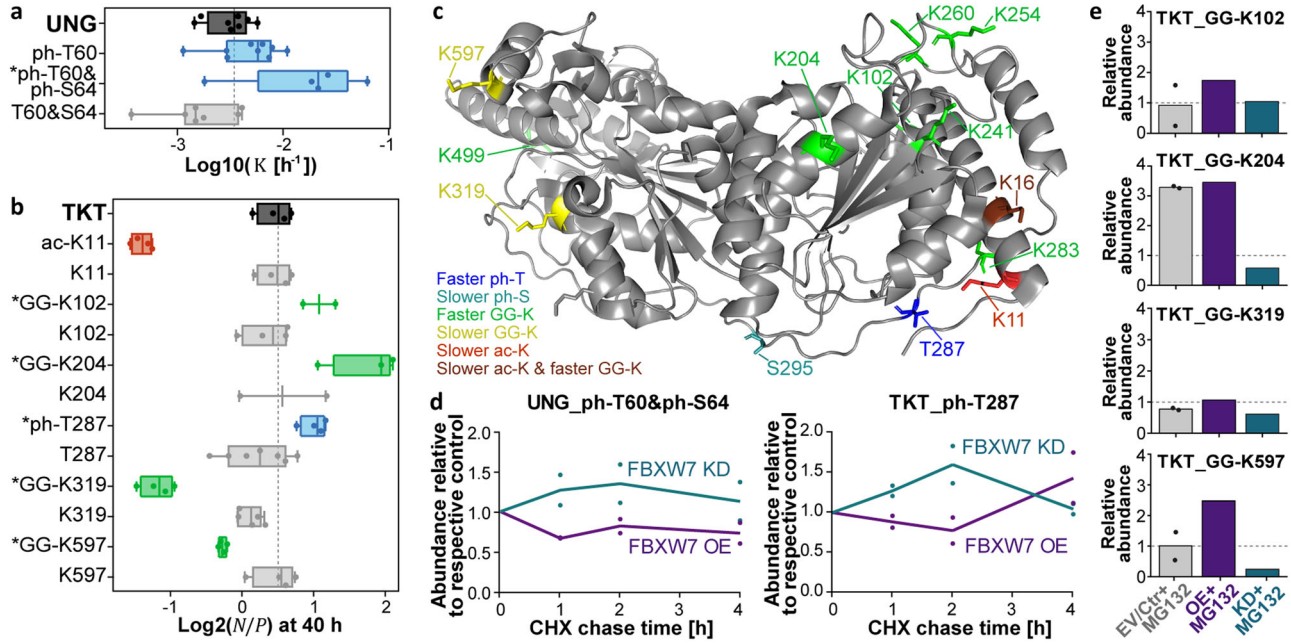

**Fig. 6 Identification of a potential F-box degron on transketolase. a** Tukey-boxplots of log10 K values for UNG and the site constituting a known FWBX7-phosphodegron (PGT(ph)PPSS(ph); box borders: 1st and 3rd quartile; lines in boxes: medians; whiskers: ranging to greatest value within 1.5× interquartile range; dots: replicates; dashed line: median of protein turnover; *: significantly different turnover to protein or unmodified counterpart). Boxplots are based on $n = 4$ cell culture replicates, except for ph-T60&S64 ($n = 3$). The number of peptides and spectra included varies for the protein and each site and replicate and can be retrieved from Supplementary Data 1. **b** Same as in panel a, but for log2 N/P ratios of TKT and selected PTMs. Boxplots are based on $n = 4$ cell culture replicates, except for GG-K204 ($n = 3$), GG-K102 and K204 (both n = 2). The number of peptides varies for sites and replicates and can be retrieved from Supplementary Data 1. The sequence around the phosphorylated threonine 287 (LAT(ph)PPQE) resembles another reported consensus sequence for FBWX7 phosphodegrons ([LIVMP]XTPXXE]). **c** TKT structure (PDB 3MOS[37]) annotated with modification sites that are linked to a significant turnover difference. Residues are colored according to their impact on the turnover determined by dSILAC. **d** Line charts showing the average relative abundance of the phosphodegron-resembling peptides of UNG (left panel) and TKT (right panel) in cycloheximide (CHX) chases after overexpression (OE) or knockdown (KD) of FBWX7. Phosphopeptide profiles were acquired via targeted parallel-reaction monitoring assays. Points represent $n = 2$ cell culture replicates. **e** Bar charts illustrating the response of ubiquitination for individual lysine sites of TKT after 2 h CHX treatment and proteasome inhibition using MG132 (EV: empty vector; CTR: control siRNA, $n = 1$ replicate). Di-Gly peptide profiles were acquired via targeted parallel-reaction monitoring assays and relative intensities were calculated using appropriate controls (EV/CTR for EV/CTR+MG132, OE for OE+MG132, CTR+MG132 for KD+MG132). Points represent respective control samples, EV and CTR ($n = 2$ biological replicates). Source data are provided as a Source Data file.

**SPOT profiling suggests a new degradation route for TKT.** Conditional phosphodegrons are well-known examples of PTM motifs that directly regulate protein turnover by recruiting ubiquitinating E3 ligases. Our analysis detected a known FBXW7-binding phosphodegron in a doubly phosphorylated peptidoform (ph-T60&ph-S64) of the glycosylase UNG[36] (Fig. 6a). We also located potential phosphodegrons on several other proteins (e.g., TKT, PNKP, DUT) that resemble the consensus binding motifs of the same F-box protein and showed an increased turnover (Supplementary Data 3). Among those, TKT, an enzyme of the pentose phosphate pathway, was highly modified, and 15 of 28 PTMs showed a significantly difference in turnover (Fig. 6b, Supplementary Fig. 9a). In addition to the potential degron around phosphorylated T287, we identified several ac-K sites exhibiting a slower turnover. Again, one of these sites clashed with a backbone carbonyl group in the crystal structure of TKT (PDB 3MOS[37]) potentially suggesting some structural change tied to acetylation (ac-K11, Supplementary Fig. 9b). Furthermore, multiple ubiquitin-remnant sites linked to faster turnover were found on one side of the protein structure, whereas two GG-K sites that showed a reduced turnover were located on the opposite side of TKT (Fig. 6c).

To further support the interpretation that phosphorylated T287 may function as a phosphodegron in TKT, we performed cycloheximide (CHX) chase experiments in HEK293T cells for up to 4 hours and quantified the relative abundance of the TKT and UNG phospho-peptidoforms after transfection with an FBXW7 expression vector, an siRNA targeting FBXW7, or respective controls. Longer chases were omitted since overexpressed FBXW7 showed a rather short half-life (Supplementary Fig. 9c). As one would expect if FBXW7 was involved in the degradation of TKT and UNG, the intensities of the corresponding phosphopeptides were increased in the knockdown compared to the overexpression samples (Fig. 6d, Supplementary Data 7). Additionally, we aimed to characterize GG-K sites on TKT by analysing their abundance in 2 and 4 h CHX chases after proteasome inhibition of the transfected cells. For ubiquitination sites that lead to the degradation of TKT and are controlled by FBWX7, proteasome inhibition should result in an increase in the controls and, likely, an even further increase when FBWX7 is overexpressed. Moreover, the knockdown of FBXW7 should decrease ubiquitination in MG132 treated cells. The levels of slower turned over GG-K319 and GG-K597 and of most of the monitored GG-K sites with faster turnover did not increase after proteasome inhibition indicating that they do not mark TKT for proteasomal degradation (Fig. 6e, Supplementary Fig. 9d, Supplementary Data 7). In contrast, GG-K204 showed the expected behavior qualifying it as a candidate site that may mediate

FBXW7-regulated, proteasomal degradation of TKT. These results exemplify that SPOT data can prioritize PTM sites for the mechanistic investigation of degradation routes of individual proteins.

## Discussion

So far, only a small number of reports have examined the association of phosphorylation and protein degradation on a proteome-wide scale[38–41]. The majority assessed phospho-specific turnover indirectly by comparing protein half-lives that were computed including or excluding all available phosphopeptides[38], by inferring degrons from responses of co-occurring phosphorylation and ubiquitination sites upon proteasome inhibition[39], or by integrating protein turnover data with counts of previously reported modifications sites[40]. Recently, Wu et al.[42] reported results of pulsed SILAC experiments in two HeLa cell lines and obtained half-lives for ~2000 pairs of 'phosphomodiforms' and their corresponding unmodified peptides in each cell line (>10,000 such pairs in the study presented here). Despite many differences in detail and coverage, their analysis revealed several parallels to our study. Wu et al. also concluded that analysing the turnover differences of peptidoform counterparts allows for more robust interpretations of the impact of PTMs than global comparisons of half-lives or turnover. Likewise, they observed heterogeneous, site-specific associations of phosphorylation and protein turnover and found that phosphorylation more often decreased than increased turnover. By comparing protein half-lives to published thermal stability data, they further concluded that phosphorylation may affect turnover related to structural features of a protein, which is in line with our analysis. Last, gene enrichment analysis also led them to suspect that the activity of enzymes may be regulated by phosphorylation associated to turnover.

The current work not only complements prior studies but goes substantially beyond by providing a very large body of steady-state turnover data for phosphorylated, acetylated, and ubiquitin-remnant peptidoforms. Moreover, SPOT profiling detected differences in protein turnover for a large number of modification sites. In contrast to previous studies, we refrain from interpreting measured differences in turnover exclusively as changes in protein stability resulting directly from a modification. Instead, we emphasize that turnover of proteins and their PTMs determined by metabolic labeling might also be influenced by other factors. Proteins or peptides that are irreversibly lost from the pool of analytes, for example by secretion or proteolytical processing, will feature an apparent higher measured turnover, but might still not be more quickly degraded compared to the unprocessed protein[7–9]. Differential turnover analyses may also identify PTMs that occur with different probabilities on newly synthesized compared to older proteins without necessarily implying a particular function within the degradation pathway for these sites. This is exemplified by the slower turned over ac-KP sites that likely just accumulate on 'aged' proteins due to reduced activity or accessibility of HDACs that would otherwise remove the modification. Little is known about phosphatase specificities, but it can be speculated that a fraction of the ph-S/TP sites with slower turnover may also result from reduced accessibility to active phosphatases.

Our analysis also uncovered modification sites that do not appear to affect protein half-life directly but have a different primary function that ultimately may still lead to a similar consequence in an indirect fashion. For example, a change in enzymatic activity, protein interactions, structure, or cellular location elicited by a particular PTM may alter the accessibility of a protein to the degradation machinery and thus influence its half-life indirectly. We also note that not all sites that show differential turnover may be functionally relevant. Some may simply co-occur with other sites that do alter protein turnover. This may still constitute useful information that standard bottom-up proteomics data typically does not provide.

Similarly, PTMs that do not show a significant change in measured turnover can still have an important function. There are many possible reasons why 80 % of the sites with known regulatory functions identified in this study showed no turnover difference. For instance, the respective PTM might not exert the annotated function in this specific cell line or cellular state (e.g., due to a missing interaction partner) or the protein state associated with the PTM might not alter the stability of the protein. Moreover, the ability to detect turnover differences depends on the stoichiometry of the modification. The higher the fraction of the protein that carries the modification, the more difficult it becomes to measure a turnover disparity between the modified peptide and the total protein. At the same time, the reliable quantification of the counterpart peptide, which would uncover the turnover difference, becomes more challenging due to its low abundance. The connection of protein turnover and modifications may also be cell type-specific. Thus, the extension of the SPOT approach to other cell types and cellular states may provide additional insights in the future.

Today, there is an enormous gap between the ease of detecting thousands of PTMs by mass spectrometry and the challenge to characterize their function within cellular processes. More specifically, >96 % of the PTMs observed in this study have no ascribed function. In this context, we envision that SPOT profiling will become a powerful tool to contribute to a better understanding of the role of PTMs in regulating protein structure and function. Our current data supports previous findings but also points to numerous PTMs with yet unknown cellular functions. These may be involved in regulating protein stability directly, indirectly or result from other factors that change the turnover of a protein. We explored a limited number of the many possible hypotheses and point out that more tailored future experiments are needed to show if and how particular PTMs are mechanistically implicated. We anticipate that the interactive website spot.proteomics.wzw.tum.de will facilitate such investigations. Clearly, the molecular complexity of cellular systems makes the interpretation of turnover changes linked to protein modifications far from trivial. In particular, an observed faster or slower turnover of a PTM peptidoform does not imply that this difference is caused by the PTM. In other words, beyond the direct involvement of the PTMs, there are several alternative scenarios that might explain the observed correlation of certain PTMs with protein turnover. Nevertheless, PTM-resolved turnover measurements can provide valuable information on PTM dynamics. Particularly, the integration of SPOT data with other layers of proteome-wide measurements, structural analyses, and drug-response data should aid the prioritization of protein modification sites for functional investigations and the current study, its data and public website provides ample opportunity for accomplishing this.

## Methods

### Cell culture

*Dynamic SILAC.* For SILAC pulse experiments, HeLa cells (ATCC, #CCL-2) were cultured in SILAC DMEM (Thermo Fisher Scientific) supplemented with 10 % dialyzed FBS (Gibco™), 1 % antibiotic, antimycotic solution (Sigma), 1.74 mM L-proline (≥ 99%, Sigma), and 4 mM L-glutamine. L-lysine and L-arginine were added to a final concentration of 0.798 mM and 0.398 mM, respectively, in either light (Lys-$^{12}C_6^{14}N_2$/K0, isotope purity ≥ 99 %; Arg-$^{12}C_6^{14}N_4$/R0 ≥ 98 %, Sigma) or heavy form (Lys-$^{13}C_6^{15}N_2$/K8, isotope purity ≥ 99 %; Arg-$^{13}C_6^{15}N_4$/R10, isotope purity ≥ 99 %, Cambridge Isotope Laboratories). Completely heavy (K8R10) labeled cells were obtained after cultivating light cells in SILAC medium containing K8 and R10 for at least 10 cell doublings. All dynamic SILAC experiments were performed in quadruplicates including two label swap experiments, i.e. twice

starting with nonlabeled (K0R0) cells and twice starting with K8R10 labeled cells. The K0R0 (or K8R10) SILAC medium on K0R0 (or K8R10) labeled cells was removed 40 h after cell seeding, cells were washed twice using phosphate-buffered saline (PBS) containing $Mg^{2+}/Ca^{2+}$ (Sigma), and K8R10 (or K0R0) SILAC medium was added. For individual pulse time points (dSILAC experiment), cells were lysed after 1 h, 6 h, 24 h, and 40 h. For the pulse time-course (dSILAC-TMT) experiment, cells were collected immediately before the pulse start and after 1, 3, 6, 10, 16, 24, 34, and 48 h. Additionally, completely K8R10 (or K0R0) labeled cells were lysed and included as "infinite h" (>200 h) time-point.

*Treatments and transfections.* To study enrichment of modified peptidoforms in different complex fractions, HeLa cells were grown in SILAC DMEM as described above. RPMI8226 cells (ATCC, #CCL-155) were grown in SILAC RPMI medium (Silantes) similarly to HeLa cells but using 0.274 mM L-lysine and 1.15 mM L-arginine. Proteasomes of K0R0 nonlabeled cells were inhibited using 1 μM MG132 or Bortezomib (Selleckchem) and labeled (K8R10) control cells were treated with 0.1 % DMSO.

For FBXW7 overexpression experiments, isoform 1 of FBXW7 was cloned into pcDNA3.1 (Life Technologies, V79020) introducing an N-terminal FLAG-Tag, and pcDNA3.1-N-FLAG was used as the empty vector. All cDNAs were sequenced. For transient transfections of the plasmid or siRNAs into HEK293T (ATCC, #CRL-3216), cells were plated one day before at 40 % confluence. The next day, transfection of pcDNA3.1-FLAG-FBXW7 or pcDNA3.1-N-FLAG plasmid DNA was performed using $CaCl_2$ in a BES (N,N-bis(2-hydroxyethyl)-2-aminoethanesulfonic acid)-buffered system. FBXW7 siRNAs were transfected using Lipofectamine RNAiMAX reagent (Invitrogen) according to the manufacture's protocol. Knockdown efficiency was assessed 2 days later. FBXW7 siRNA (M-004264-02) and siCTRL/control siRNA (D-001206-13) were obtained from Dharmacon. Cycloheximide, an inhibitor of eukaryotic mRNA translation, was used to examine protein degradation. A stock solution of 100 mg/ml Cycloheximide (Sigma) in 100 % ethanol was prepared right before use and cells were treated with 100 μg/ml for 0, 1, 2, and 4 h. Transiently transfected HEK293T cells were treated 24 (plasmids) and 48 h (siRNA) after transfection. To inhibit proteasomes, 10 μM MG132 (Tocris) was added to the cycloheximide treatment when indicated. All nucleotide sequences are listed in Supplementary Data 8.

**Cell lysis and protein digestion.** Cells from dynamic SILAC and CHX chase experiments were lysed in 8 M urea in 40 mM Tris-HCl, pH 7.6 containing protease inhibitor (cOmplete™ Mini, Roche), phosphatase inhibitor cocktails (prepared in-house according to the formula of Phosphatase Inhibitor Cocktail 1, 2, and 3 from Sigma), lysine deacetylase, deubiquitinase, and proteasome inhibitors (5 mM sodium butyrate, 50 μM PR619, 50 μM MG132). Lysates were cleared for 20 min at 20,000 g and 4 °C, and protein concentrations were determined with the Pierce™ Coomassie Protein Assay Kit (Thermo Scientific). After reduction (10 mM DTT, 30 min at 30 °C) and alkylation (50 mM chloroacetamide, 30 min at room temperature), lysates were diluted to <1.6 M urea using 40 mM Tris-HCl (pH 7.6). Digestion was performed by adding trypsin (Promega, 1:50 enzyme-to-substrate ratio) and incubating overnight at 37 °C and 700 rpm. Digests were acidified to pH < 3 by addition of neat formic acid (FA) or trifluoroacetic acid (TFA) to 1 % or 0.5 % and centrifuged to pellet insoluble matter. Supernatants were desalted using tC18, reverse-phase (RP) solid-phase extraction (SPE) cartridges (Waters Corp.; wash solvent: 0.1 % FA or 0.07 % TFA; elution solvent: 0.1 % FA or 0.07 % TFA in 50 % acetonitrile (ACN)). For phosphopeptide enrichment, eluates were adjusted to a final ACN concentration of 30 % by the addition of 0.7 % TFA. Otherwise, peptides were dried by vacuum centrifugation.

**TMT labeling.** TMT labeling of 100 μg cleaned-up peptides (corresponding to initial quantity of 200 μg protein) from the pulse time-series was performed using TMT10plex reagents (Thermo Scientific) according to our recently published protocol[43]. In brief, 100 μg of TMT reagent in 5 μl of anhydrous ACN was used to label peptides in 20 μl of 50 mM Hepes buffer, pH 8.5, and the labeling reaction was stopped after 1 h of incubation by adding 3 μl of 5 % hydroxylamine. After pooling and vacuum drying, peptides were dissolved in 0.07 % trifluoroacetic acid (TFA) and cleaned up using RP-SPE cartridges (wash solvent: 0.07 % TFA; elution solvent: 0.07 % FA in 50 % ACN). Eluates were adjusted to a final ACN concentration of 30 % by the addition of 0.7 % TFA for subsequent phosphopeptide enrichment.

**Enrichment of phosphorylated peptides.** Immobilized metal ion affinity chromatography (IMAC) for the enrichment of phosphopeptides in dynamic SILAC (dSILAC) samples was performed using a ProPac IMAC 10 column (4 × 50 mm, Thermo Fisher Scientific) charged with iron(III) ions. Dynamic SILAC (1.4 mg peptides) and pooled dSILAC-TMT samples (1 mg peptides) were dissolved in 0.5 ml IMAC loading solvent (0.07 % TFA in 30 % ACN), loaded onto the column connected to an Aekta HPLC system (GE Healthcare Life Sciences), and washed for 5 min at 0.2 ml/min. Meanwhile, the flow-through containing non-phosphopeptides was collected for subsequent enrichment of di-glycine- and acetyl-lysine. Bound phosphopeptides were eluted and collected applying a two-step gradient from 0 % to 12 % IMAC elution solvent (0.315 % $NH_4OH$) in 1.5 min at 0.55 ml/min and to 26 %

elution solvent in 5 min at 3 ml/min. Flow-through and eluted peptides were dried by vacuum centrifugation and stored at −80 °C until fractionation.

Phosphopeptides in samples of CHX chase experiments (~100 μg each) were enriched using Fe(III)-NTA cartridges and the standard manufacturer protocol on the AssayMAP Bravo Platform (Agilent Technologies). In brief, peptides were loaded in 200 μl loading solvent (0.1 % TFA in 80 % ACN). The cartridges were washed with 50 μl loading solvent, and phosphopeptides were eluted using 60 μl of 1 % $NH_4OH$. Phosphopeptides were dried down and stored at −20 °C until LC-MS analysis.

**Immunoaffinity purification of di-Gly and acetylated peptides.** Ubiquitin-remnant (di-glycine) and acetylated peptides were enriched via immunoaffinity purification (IAP) using commercial antibody beads (PTMScan® Ubiquitin Remnant Motif (K-ε-GG) Kit #5562, PTMScan® Acetyl-Lysine Motif [Ac-K] Kit #13416, Cell Signaling Technology) according to manufacturer instructions with some adjustments. Before use, Ubiquitin-remnant antibodies were cross-linked to agarose beads. Following three washes of UbiScan beads with 100 mM sodium tetraborate decahydrate (Borax), pH 8.8, cross-linking was carried out at RT for 30 min with end-over-end rotation using freshly prepared 20 mM dimethyl pimelimidate in 100 mM Borax. The reaction was stopped with 200 mM ethanolamine, pH 8.0 (2 washes, then incubation at 4 °C for 2 h with end-over-end rotation). Beads were washed three times with IAP buffer (50 mM MOPS, pH 7.2, 10 mM $Na_2HPO_4$, 50 mM NaCl), re-suspended in PBS containing 0.02 % (w/v) sodium azide, and stored at 4 °C.

For enrichment, dried peptides (~1.4 mg peptides for dSILAC samples, ~1 mg peptides for dSILAC-TMT samples, ~0.5 mg for CHX chase samples) were reconstituted in 1 ml IAP buffer and centrifuged at 20,000 g for 10 min at 4 °C to pellet insoluble matter. Supernatants were incubated with the antibody bead slurry (10 μl settled beads) for 1 h at 4 °C with end-over-end rotation. After 2 washes with IAP buffer (only 1 wash for acetyl beads) and 2 washes with PBS, modified peptides were eluted at RT with 2 × 50 μl of 0.15 % TFA for 5 min. Eluates were desalted using self-packed StageTips (three disks of C18 material, Ø 1.5 mm, 3 M Empore™; wash solvent: 0.1 % FA; elution solvent: 0.1 % FA in 40 % ACN). Peptides were dried by vacuum centrifugation and stored at −20 °C until LC-MS analysis.

**Peptide fractionation.** For hydrophilic strong anion exchange (hSAX) chromatography of TMT-labeled whole proteome samples, 50 μg cleaned-up peptides were reconstituted in hSAX solvent A (5 mM Tris-HCl, pH 8.5) and loaded onto an IonPac AS24 strong anion exchange column (2 × 250 mm) equipped with an IonPac AG24 guard column (2 × 50 mm) and coupled to a Dionex Ultimate 3000 HPLC system (Thermo Fisher Scientific). Samples were washed using 100 % hSAX solvent A for 3 min at 250 μl/min and subsequently eluted applying a two-step gradient from 0 to 25 hSAX solvent B (1 M NaCl in 5 mM Tris-HCl, pH 8.5) in 24 min and to 100 % in 13 min. Forty fractions (1 minute each) were collected and acidified with 5 μl neat FA. Early and late fractions were pooled (1–4, 5–7, 8–9, 26–27, 28–30, 31–33, 34–35, 36–40), and the resulting 24 fractions were desalted using self-packed StageTips (three disks of C18 material; wash solvent: 0.1 % FA; elution solvent: 0.1 % FA in 50 % ACN). Eluted fractions were frozen, dried by vacuum centrifugation, and stored at −20 °C until LC-MS analysis.

Basic RP tip fractionation of whole proteome dSILAC and phosphopeptide samples was performed as described previously[44] with minor modifications. In brief, dried peptides were first cleaned-up using self-packed StageTips (five disks of C18 material, wash solvent: 0.1 % FA) without elution. Instead, peptides bound to StageTips were re-buffered using 50 μl of 25 mM $NH_4COOH$ and then sequentially eluted using 50 μl of 25 mm $NH_4COOH$ containing increasing concentrations of ACN (scheme 1 for phosphopeptides from dSILAC-TMT and unmodified peptides from dSILAC experiments: 5, 7.5, 10, 12.5, 15, 17.5, and 50 % ACN; scheme 2 for phosphopeptides from dSILAC experiments: 2.5, 7.5, 12.5, and 50 %). High and low ACN fractions were combined to obtain 6 or 4 fractions (scheme 1: flow-through + 17.5 % fraction, 5 % + 50 % fraction; scheme 2: flow-through + 50 %). Fractions were dried by vacuum centrifugation and stored at −20 °C until LC-MS measurement.

**Size Exclusion Chromatography.** For size exclusion chromatography (SEC), Hela and RPMI8226 cells were lysed in ice-cold SEC lysis buffer (50 mM Tris-HCl, pH 7.4, 100 mM NaCl, 10 % glycerol, 0.1 % Triton-X-100, protease and phosphatase inhibitor cocktails, sodium butyrate, PR619, and MG132). After passing through a 0.2 μm filter, 1 mg of protein in 0.25 ml was injected on a Superose 6 10/30 GL column connected to an Äkta pure 25 (GE Healthcare) and separated at 4 °C using SEC buffer (50 mM Tris-HCl, pH 7.4, 100 mM NaCl, 10 % glycerol) at a flow rate of 250 μl/min. Molecular mass calibration was performed with the high molecular mass gel filtration calibration kit (GE Healthcare). After the void volume, 48 fractions of 0.31 ml each were collected, vacuum-dried, and re-solubilized in 80 μl urea digestion buffer. Digestion using 0.5 μg trypsin per fraction and sample desalting with self-packed StageTips was performed as described above.

**Immunoblotting.** For immunoblot analyses, cells were lysed in NP-40 buffer (NaCl 150 mM, Tris-HCl 50 mM, $MgCl_2$ 5 mM, EDTA 1 mM, 0.1 % NP-40, 5 % glycerol,

protease/phosphatase inhibitors), incubated for 20 min on ice, and cleared by centrifugation. Subsequent immunoblotting was performed as previously described[45]. Membranes were blocked in blocking buffer (5 % nonfat dry milk in PBS with 0.1 % Tween-20) and incubated with primary antibodies, which were diluted in blocking buffer, at 4 °C overnight. Detection was performed using horseradish peroxidase-conjugated secondary antibodies (NAV931V and NAV934V; GE Healthcare, 1:10000). The following primary antibodies were used at the indicated dilutions: anti-FLAG (Sigma; Clone-M2: #F3165, 1:1000), anti-TKT (#sc-390179, Santa Cruz Biotechnologies, 1:3000), anti-GAPDH (#sc-47724, Santa Cruz Biotechnologies, 1:1000), and anti-PLK1 (#33-1700, Thermo Fisher Scientific; 1:500).

## Mass spectrometry data acquisition

*Data-dependent acquisition of dynamic SILAC samples.* LC-MS measurements of dSILAC samples were performed with an Ultimate 3000 RSLCnano system coupled to a Q-Exactive HF-X mass spectrometer (Thermo Fisher Scientific). Basic RP-fractionated peptides were reconstituted in 0.5 % FA and all other samples in 0.1 % FA. An amount corresponding to ~0.5 µg peptides of whole proteome fractions, one-fifth of all phosphopeptide fractions, and half of the acetyl and di-glycine IAPs were injected onto a trap column (75 µm x 2 cm, packed in-house with 5 µm C18 resin; Reprosil PUR AQ, Dr. Maisch), washed with 0.1 % FA for 10 min at a flow rate of 5 µl/min and subsequently transferred to an analytical column (75 µm x 45 cm, packed in-house with 3 µm C18 resin; Reprosil Gold, Dr. Maisch). Peptides were separated at 300 nl/min using a 50 min linear gradient from 4 to 32 % LC solvent B (0.1 % FA, 5 % DMSO in ACN) in LC solvent A (0.1 % FA in 5 % DMSO), except for phosphopeptides which were separated in a two-step gradient from 2 to 15 to 27 % solvent B. The HF-X was operated in positive ionization and data-dependent acquisition (DDA) mode. MS1 spectra were recorded in the Orbitrap from 360 to 1300 m/z at a resolution of 60 K and using an automatic gain control (AGC) target value of 3e6 charges and a maximum injection time of 25 ms. Up to 25, 15, and 12 of the most abundant precursors (topN) were selected for HCD fragmentation at 26 % normalized collision energy (NCE) for nonmodified, phosphorylated, and acetylated/di-glycine peptides, respectively. MS2 spectra were acquired at 15 K resolution using an isolation window of 1.3 m/z, an AGC target value of 1e5 charges (2e5 charges for phosphopeptides), and a maxIT of 100 ms (22 ms for whole proteome fractions). The first mass was fixed to 100 m/z and dynamic exclusion was set to 25 s.

*Data-dependent acquisition of dSILAC-TMT samples.* LC-MS measurements of dSILAC-TMT samples were performed similarly as described above but using a Fusion Lumos Tribrid mass spectrometer (Thermo Fisher Scientific) in MS3 mode with the following changes: One-third of the phosphoproteome fractions and ~0.7 µg peptides of whole proteome fractions were injected. Peptides were separated using a 100 min linear gradient from 4 % to 32 % LC solvent B. MS1 spectra were recorded using an AGC target of 4e5 charges and a maxIT of 50 ms. MS2 spectra for peptide identification were recorded in the ion trap in rapid scan mode (isolation window 0.7 m/z, AGC target value of 2e4, maxIT of 100 ms, dynamic exclusion of 90 s). For unmodified, di-glycine, and acetylated peptides up to 10 precursors were fragmented via CID (NCE of 35 %, activation Q of 0.25), whereas phosphopeptides were fragmented using HCD (NCE of 33 %) and a cycle time of 2 s. Subsequently, an additional MS3 spectrum for TMT quantification was obtained for each peptide precursor in the Orbitrap at 60K resolution (scan range 100–1000 m/z, charge-dependent isolation window from 1.3 (2+) to 0.7 (5–6+) m/z, AGC of 1.2e5 charges, maxIT of 110 ms). For this, the precursor was fragmented again as for MS2 analysis, followed by the synchronous selection of the 10 most intense peptide fragments in the ion trap and further fragmentation via HCD using an NCE of 55 %.

*Data-dependent acquisition of SEC samples.* Whole proteome profiles of SEC fractions were acquired using micro-flow conditions on a Dionex UltiMate 3000 RSLCnano System equipped with a Vanquish pump module and coupled to a Q-Exactive HF-X mass spectrometer as described previously[46]. In brief, ~3.5 µg peptides per fraction were injected directly onto a commercially available Acclaim PepMap 100 C18 LC column (2 µm particle size, 1 mm ID × 150 mm; Thermo Fisher Scientific). Peptides were separated at a flow rate of 50 µl/min using a 15 min linear gradient of 3 to 28 % micro-flow solvent B (0.1 % FA and 3 % DMSO in ACN) in micro-flow solvent A (0.1 % FA and 3 % DMSO). MS data acquisition was performed as described for whole proteomes of dSILAC samples with minor adjustments: MS1 maxIT was set to 50 ms, the 12 most abundant precursors were fragmented at 28 % NCE, and dynamic exclusion was 10 s.

*Parallel-reaction-monitoring in SEC and CHX chase samples.* For measurements of differentially modified proteasomal and ribosomal peptidoforms, each three adjacent SEC fractions were pooled to reduce the number of fractions to 16, and ~0.3 µg per fraction was injected into the LC-MS system. For parallel-reaction-monitoring (PRM) analyses of phosphorylated and ubiquitinated peptides in CHX chase samples, half of the phospho-enriched samples and the di-glycine enrichments were used. Additionally, PROCAL peptide standard (JPT Peptide Technologies) was spiked into all samples (90–175 fmol/injection). Similar LC-MS settings were used as described above for dSILAC samples, but the HF-X was operated in PRM mode with the following adjustments: Full MS scans were recorded at 45K resolution using a maxIT of 20 ms. Targeted MS2 spectra were acquired at 60K resolution and using an AGC target value of 1e6 charges. The maxIT was adjusted according to the number of targeted precursors at any retention time (160 ms for SEC fractions, 180 ms for CHX chase samples) to obtain a maximal cycle time of ~2 s. For SEC fractions, several survey runs were performed to narrow down the number of detectable proteasomal and ribosomal peptides. In the end, eight proteasomal and two ribosomal peptides were scheduled for PRM in their respective light and heavy SILAC state in the 15 SEC fractions. In CHX chase samples, nine phosphodegrons of five proteins and nine ubiquitinated peptides of TKT were targeted. Additionally, six PROCAL peptides, which were distributed across the whole retention time range, were also part of the inclusion list for all PRM runs.

## Data processing and analysis

*Database searching.* MaxQuant v1.6.0.16 with its built-in search engine Andromeda was used to identify and quantify peptides for DDA type of experiments. Tandem mass spectra were searched against the human Swiss-Prot database (42,271 entries including splice variants, downloaded on Feb 2018) supplemented with common contaminants. Unless stated otherwise, MaxQuant's default parameters were applied. These included for all searches: Trypsin/P as the proteolytic enzyme with up to two missed cleavage sites allowed; carbamidomethylation of cysteine as fixed modification, oxidation of methionine and N-terminal protein acetylation as variable modifications; Andromeda score and delta score cut-offs for modified peptides of 40 and 6, respectively; 1 % peptide spectrum match (PSM) and protein false discover rate (FDR) employing a target-decoy approach using reversed protein sequences. The precursor tolerance was set to ±4.5 ppm and fragment ion tolerance to ±20 ppm (FTMS) and 0.4 Da (ITMS). Phosphorylation on serine, threonine, and tyrosine, or acetylation, or di-glycine on lysine were allowed as variable modifications specifically for the corresponding sub-proteome enrichments. For dSILAC and SEC samples, Lys0/Arg0 and Lys8/Arg10 were specified as metabolic labels. For dSILAC-TMT samples, TMT10 was specified as a label within a reporter ion MS3 experiment type, and Lys8 and Arg10 were set as additional variable modifications. Isotope impurities of the TMT batch were specified in the configuration of TMT modifications to allow MaxQuant the automated correction of TMT intensities. The match-between-runs option was enabled for all non-TMT-labeled samples.

*Processing of single time-point dynamic SILAC data.* Hits to the decoy database and potential nonhuman contaminants were removed from the dSILAC dataset. Ratios of newly synthesized to pre-existing ($N/P$) peptides and proteins were calculated based on heavy-to-light ($H/L$) ratios specified in the MaxQuant's evidence.txt output file. $H/L$ ratios were log-transformed and the median of all evidence entries matching to the same peptide sequences or protein group was calculated. Oxidized and nonoxidized versions of the same peptidoform were combined and only protein group unique peptides were included to compute protein $H/L$ ratios. For replicates for which the cell culture medium was switched from K0R0 to K8R10, $H/L$ ratios corresponded to $N/P$ ratios, while the inverse value of $H/L$ ratio was used as $N/P$ ratio for replicates starting with K8R10 labeled cells.

*Processing of time-course dynamic SILAC-TMT data.* TMT intensities were extracted from the evidence.txt output file, which discriminates K0/R0 and the K8/R10 labeled peptides, and utilized for all quantitative analyses. Entries mapping to reverse database hits or nonhuman contaminants or such that could not be assigned to either of both turnover types (i.e., missed cleaved peptides carrying both a light and a heavy version of lysine or arginine, C-terminal peptides without any lysine or arginine residue) were removed for further analyses.

TMT intensities in dSILAC-TMT experiments are expected to show a constantly increasing or decreasing behavior reflecting label incorporation or loss, respectively. Consequently, co-isolation and fragmentation of peptides with opposing quantitative characteristics can result in severe ratio distortion and adulteration of turnover rate estimations or failure to pass criteria for the fitting of turnover curves. To tackle this quantitative bias, the average label incorporation curve was subtracted from all label loss curves, and the average label loss curve was subtracted from all label incorporation curves for all sub-proteome samples. This was implemented with the help of the outermost TMT channels that, owing to the experimental design, indicated the degree of co-isolation. More precisely, for adjustment of label loss curves, TMT intensities were summed up per channel and for all evidence entries that represented label incorporation and exhibited less than 10 % ratio compression (calculated by the ratio of the 0 h to the infinite h TMT channel). Summed intensities were divided by the summed intensity of the last TMT channel to obtain the normalized average label incorporation curve as fractions of average label incorporation per time-point ($F_{av\_incorp}(t)$). Subsequently, the fractions of intensity corresponding to the average synthesis curve were subtracted from all reporter intensities of evidence entries resembling label loss ($RI_{loss}(t)$) to obtain their corrected reporter intensities ($RI\_corr_{loss}$, see also Supplementary Fig. 2b for a visualization):

$$RI\_corr_{loss}(t) = RI_{loss}(t) - RI_{loss}(\infty) \cdot F_{av\_incorp}(t) \qquad (1)$$

TMT reporter intensities of evidence entries reflecting label incorporation were corrected accordingly by utilizing the fractions of average label loss ($F_{av\_loss}(t)$):

$$RI\_corr_{incorp}(t) = RI_{incorp}(t) - RI_{incorp}(0) \cdot F_{av\_loss}(t) \quad (2)$$

After subtraction of co-isolated intensities, TMT reporter intensities were normalized for mixing errors.

Normalization of dSILAC-TMT data was conducted under the assumption that the total protein amount (i.e., light plus heavy labeled protein) is equal across time points. For calculation of normalization factors, only peptidoforms quantified in both SILAC states and in at least 7 TMT channels without any zero intensities in between and not mapping to human contaminants were considered. Normalization was performed in two steps (for a more detailed description see Zecha et al.[7]): First, a so-called row-wise normalization was employed. This equalized intensities of the 0 h channel of peptides resembling label loss and the 'infinite' h channel of peptides representing label incorporation and thus adjusted the differences in overall TMT intensity levels for corresponding light and heavy peptidoforms. Second, 10 channel normalization factors were computed in a column-wise normalization step that brought the total sums of TMT channels (including both light and heavy peptides) to the same level and consequently compensated for mixing errors during the multiplexing process. Subsequently, the channel normalization factors were applied to all peptidoforms.

The abundance of individual (modified) proteins may change over the time course of a dSILAC experiment violating the kinetic model and steady-state assumptions underlying the computation of turnover rates from fitted curves[7]. Therefore, reporter intensity ratios to which turnover curves were fitted were corrected for such abundance fluctuations to improve turnover estimations. The assessment of abundance changes was based on the premise that the sum of the MS1 intensities of the light and heavy version of a peptidoform at a certain time-point corresponds to its total abundance at this respective time-point. Consequently, TMT ratios could only be corrected for peptidoforms for which MS1 intensity information of both label states was available. To improve robustness and avoid artefacts during computational adjustment for abundance changes, the following additional criteria had to be fulfilled for peptides to be included: at least four quantified MS1 peak elution profiles in any replicate or two quantified MS1 peak elution profiles in at least two replicates and valid TMT intensity in at least seven TMT channels without any zero intensities in between. MS1 intensities and (separately) TMT intensities were summed up for identical peptide sequences in the same SILAC labeling state and replicate. Then, the fraction of the MS1 intensity corresponding to a peptidoform at a certain time-point was calculated for both SILAC labeling states:

$$MS\ intensity\ fraction_{loss}(t) = \frac{RI\_corr_{loss}(t)}{\sum_{all\ time-points} RI\_corr_{loss}} \cdot MS1\ intensity_{loss} \quad (3)$$

$$MS\ intensity\ fraction_{incorp}(t) = \frac{RI\_corr_{incorp}(t)}{\sum_{all\ time-points} RI\_corr_{incorp}} \cdot MS1\ intensity_{incorp} \quad (4)$$

The sum of the MS intensity fractions of label loss and incorporation was utilized as a proxy for the abundance at a certain time-point in a certain replicate:

$$Abundance(t) = MS\ intensity\ fraction_{loss}(t) + MS\ intensity\ fraction_{incorp}(t) \quad (5)$$

Factors (A) for the adjustment of intensity ratios were calculated as the abundance changes relative to the first and the last channel, respectively:

$$A_{loss}(t) = \frac{Abundance(t)}{Abundance(0)} \quad (6)$$

$$A_{incorp}(t) = \frac{Abundance(t)}{Abundance(\infty)} \quad (7)$$

The median of these factors across replicates was utilized to obtain abundance adjusted intensity ratios (R_adjust):

$$R\_adjust_{loss}(t) = \frac{RI\_corr_{loss}(t)}{median(A_{loss}(t)) \cdot RI\_corr_{loss}(0)} \quad (8)$$

$$R\_adjust_{incorp}(t) = \frac{RI\_corr_{incorp}(t)}{median(A_{incorp}(t)) \cdot RI\_corr_{incorp}(\infty)} \quad (9)$$

The kinetic model underlying the estimation of turnover rates is described in detail in our recent publication[7]. In brief, turnover (labeling) rates (K), curve maxima (A), and offsets (B) were obtained for each evidence entry via performing a nonlinear least square (NLS) optimization in R (v3.4.1, function nls) using the following equations:

$$R_{loss}(t) = (A - B) \cdot e^{-K \cdot t} + B \quad (10)$$

$$R_{incorp}(t) = (B - A) \cdot e^{-K \cdot t} + A \quad (11)$$

Resulting curve fits were filtered according to the following *KRAB* criteria: $K : 0 - 5; R^2 \geq 0.7; A : 0.7 - 1.4; B : -0.15 - 0.25$. Turnover rates of peptidoforms and proteins were obtained from a combined fit of all evidence entries filtered for *KRAB* criteria and belonging to the respective modified peptide or protein

sequence. Oxidized and nonoxidized versions were combined for computing turnover rates of peptidoforms and only protein group unique peptides were included for the determination of whole protein (group) turnover.

*Estimation of amino acid recycling.* The degree of amino acid recycling in different pulse time points was estimated harnessing quantitative information on missed cleaved peptides that were quantified in the label state representing label incorporation and the mixed labeling state (containing both a light and a heavy version of lysine and/or arginine residues) in the dSILAC-TMT dataset. First, the fractions of the MS1 intensities were calculated for label incorporation and mixed-label peptides for each time-point according to Eqs. 3 and 4. Second, the recycled fraction at certain pulse time points was calculated as the ratio of the MS1 intensity fraction of the mixed-label peptide to the sum of MS1 intensity fractions of the mixed and the label incorporation-resembling peptide:

$$Recycled\ fraction(t) = \frac{MS\ intensity\ fraction_{mixed}(t)}{MS\ intensity\ fraction_{mixed}(t) + MS\ intensity\ fraction_{incorp}(t)} \quad (12)$$

Since it was impossible to estimate and remove ratio compression for mixed-label peptides, non-adjusted TMT reporter intensities were used for the computation of the recycled fraction.

*Analysis of parallel-reaction-monitoring assays.* Skyline v20.2.0.286 was used to build spectral libraries from dSILAC DDA data processed with MaxQuant. Additionally, we predicted a spectral library for PROCAL peptides using the Prosit 2019 algorithm[47]. The raw PRM data were imported into Skyline, transitions were automatically chosen from the spectral library, and peaks were integrated using its automatic peak finding function followed by manual curation of all peak boundaries and transitions. If necessary, we manually adjusted integration boundaries and removed strongly interfered transitions meanwhile keeping at least five transitions per peptide (exception: four transitions for the ubiquitinated peptide SGK(GG)PAELLK representing GG-K597 in TKT). Only peptides that showed robust elution profiles with a dot product larger than 0.9 in at least one labeling state and condition were kept (exception: 0.89 for the acetylated peptide VSTAVLSITAK(ac)AK representing ac-K838 in PSMD1). Final transitions can be examined on the Panorama Public website (https://panoramaweb.org/SPOT.url). The summed areas under the fragment ion traces were exported to Microsoft Excel for further data processing. If elution profiles had a dot product of less than 0.6 in a certain sample, we set their residual intensities to zero.

For the analysis of SEC fractions, we first corrected PRM and DDA intensities from high-flow measurements for SILAC mixing errors. For this, an aliquot of the mixed sample that was taken before size exclusion chromatography, measured in DDA mode, and analyzed using MaxQuant. The summed K0R0 and K8R10 peptide intensities in this single shot were utilized to compute normalization factors, which were then applied to DDA and PRM measurements of SEC fractions. For CHX chase samples, we normalized different injection amounts based on intensities of retention time peptides. Further, different biological backgrounds and measurements on different MS machines led to systematic differences in intensity levels for different replicates and FBXW7 knockdown and overexpression conditions. To correct for these batch effects, we applied a normalization factor that set the phosphopeptide intensities in the 0 h CHX chase time-point to the same level in all replicates and in knockdown and overexpression samples. Subsequently, we calculated intensity ratios to respective controls (empty vector or control siRNA).

*Site-counterpart mapping and functional annotation.* For counterpart analyses, nonmodified peptidoforms were mapped to modified peptidoforms based on site information. Hence, a peptide was defined as counterpart as long as it contained the unmodified amino acid that was identified with a modification in another peptide potentially resulting in several different counterpart peptide sequences for one modified peptidoform. Modifications were not filtered for localization probability but were always automatically assigned to the most likely site.

Functional annotations from UniProt, Corum, Prosite, and PhosphoSitePlus were performed based on the first UniProt identifier for each protein group or peptide sequence. Human Protein Atlas information was only available on the gene-level and was therefore matched based on the first gene name. Enzyme-substrate information[31,32] were first matched based on the first IPI/UniProt identifier and second, for still unmatched data or for data for which no protein identifier was available, on the first gene name entry for each protein group or peptidoform. Acetylation sites were identified as targets of KDACs and KATs when inhibitor treatments or enzyme transfections have been reported to induce at least a twofold change in PTM abundance. Degrons[48] were annotated based on (modified) peptide sequences. Crystal structures of the proteasome, ENO1, and TKT were obtained from the RCSB protein databank website.

*Statistical analysis.* The Perseus software suite (v1.6.2.3) was utilized to perform correlation analyses, two-sided Student's t tests, and Fisher's exact enrichment analyses. Student's t test required at least two N/P ratios or turnover rates per group. For the comparison of peptidoform and protein turnover, only proteins with at least 3 quantified peptidoforms were included. For the comparison of

modified peptidoforms to counterparts, all possible pairs were allowed in the analyses resulting in several pairwise comparisons for modified peptides for which more than one nonmodified counterpart peptide was quantified. S0 for $t$ tests was computed in R (v3.4.1 or 3.6.0, function samr) for each statistical group comparison separately. This constant adjusts the significance cut-off of statistical analyses on the fold-change level, while accounting for differing variances across the range of measured values and groups. For further analyses, a peptidoform was defined to feature significantly slower or faster turnover when at least one of the statistical tests identified it with significantly slower or faster turnover. Peptidoforms or sites that were found with opposing regulations in different datasets or on different peptidoforms (ambiguous) were discarded. The background for Fisher's exact tests was defined by all peptidoforms that fulfilled the criteria for $t$ tests. All statistical tests were corrected for multiple testing applying a permutation-based or Benjamini-Hochberg FDR calculation at 1 % or 5 % as indicated.

Motif enrichment analyses were performed using the motifX algorithm[28] via the R package rmotifx. Sequence windows with 11 amino acids around sites that showed either significantly faster or slower turnover were specified as foreground, and sequence windows of all other identified sites with quantified turnover were used as background. Duplicates were removed, a minimum of 20 sequences was required per motif, and the significance cut-off was set to 2.5e-4 corresponding to an actual alpha-value of <0.05 after Bonferroni correction. Amino acid distributions in sequence windows of sites with significantly slower or faster turnover were plotted using pLogo[29] (https://plogo.uconn.edu/), a motif visualization algorithm in which residue heights are scaled relative to their statistical significance. Protein crystal structures were visualized in PyMOL v2.4.1 and all other data using R v3.4.1 or v3.6.0, GraphPad Prism 5, Microsoft Excel 2013, and Venn Diagram Plotter v1.5.

**Reporting summary**. Further information on research design is available in the Nature Research Reporting Summary linked to this article.

## Data availability

The MS proteomics raw data and MaxQuant search results including the fasta file from the Swiss-Prot database have been deposited to the ProteomeXchange Consortium via the PRIDE partner repository under the accession codes PXD023218 (dSILAC, dSILAC-TMT, and SEC experiments) and PXD023234 (SEC and CHX PRM assays). Raw data and results of the PRM assays can also be examined on the Panorama Public website at https://panoramaweb.org/SPOT.url (SEC and CHX experiments). Normalized and further processed data such as turnover rates and N/P ratios and nucleotide sequences are provided in the Supplementary Data files. In addition, all analysed data of modified and counterpart sites can be viewed and explored via spot.proteomics.wzw.tum.de. Source data of all figures are provided with this paper. Previously published potein crystal structures were obtained from the RCSB protein databank website (PDB IDs: 5LE5 [https://doi.org/10.2210/pdb5LE5/pdb], 3B97 [https://doi.org/10.2210/pdb3B97/pdb], 3MOS [https://doi.org/10.2210/pdb3MOS/pdb]). Source data are provided with this paper.

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

## Acknowledgements
The authors wish to thank Guillaume Medárd, Jakob Trendel, Karl Kramer, Polina Prokofeva, and Stephanie Heinzlmeir for stimulating discussions and helpful advice. The authors are also grateful to Daniel Zolg, Miriam Abele, Christina Ludwig, Franziska Hackbarth, Chien-Yun Lee, and Andreas Zellner for maintaining and operating the mass spectrometers. Further, the authors thankfully acknowledge Daniel Zolg and Patroklos Samaras for their help on generating spectral libraries and Christina Ludwig for her advice on the Panorama Upload. This work was partly funded by the German Cancer Research Center (DKFZ), the European Research Council (ERC Advanced Grant TOPAS # 833710 (B.K., J.Z.), ERC Consolidator Grant BCM-UPS #682473 (F.B.)), the Deutsche Forschungsgemeinschaft (SFB1321 (B.K., J.Z.), SFB 1309 (B.K., Y.C.), SFB924 (B.K., J.M.), SFB1335 (F.B.)), and the German Federal Ministry of Education and Research (BMBF, grant #031L0168 (BK, M.W., W.G.)).

## Author contributions
B.K. and J.Z conceived the project and designed the research. J.Z., R.S., and Y.C. carried out the experimental work with crucial help from J.M. J.Z. developed the turnover analysis strategy, processed MS data, and performed the bioinformatic workup. W.G. developed the Shiny application with crucial help and input from M.W., J.Z., J.M., and B.K. F.B. provided important conceptual advice. J.Z. generated figures and wrote the manuscript. B.K. and J.Z. revised the manuscript with input from R.S. and F.B.

## Funding

## Competing interests
M.W. and B.K. are founders and shareholders of OmicScouts GmbH and MSAID GmbH. They have no operational role in either company. J.Z. is now an employee of AstraZeneca. The remaining authors do not declare any competing interest.
