## [Peer Review File · Nature Communications]

REVIEWERS' COMMENTS

Reviewer #1 (Remarks to the Author):

The authors have fully addressed all the points raised, and the manuscript can be recommended for publication.

Reviewer #2 (Remarks to the Author):

The authors have done a good job of addressing the concerns highlighted by the reviewers, particularly the issues around correlation and causation.

While I was able to access the web portal (a great feature for interrogation of this dataset), it would be ideal if the authors were able to add a protein level search function in addition to the dropdown menu, allowing interrogation based on e.g. protein name.

Reviewer #3 (Remarks to the Author):

I appreciate the detailed response by the authors. However, I do not think their revised manuscript adequately addresses the points I raised. I had three major points: the confusion of correlation and causation, the lack of validation and the fact that the method is not really new. The novelty of a method is less critical for a non-method journal, so I am only focussing on the other two points.

1. Correlation and causation:  The authors still do not seem to see the fundamental flaw in the way they interpret their data. They say that they do not agree with my notion that the paper contains no data to support the view that Ubi and/or Ac can affect protein stability. In their point-by-

point response they write: “While we may not provide formal proof regarding how this happens, the observation that e. g. acetylation is more often associated with slower than faster turnover implies an involvement of the PTM (regardless of cause or consequence).”

This statement is wrong and highlights a fundamental misconception in how the authors interpret the data. Maybe this becomes clearer by these two alternative interpretations that come to my mind:

A) Acetylation could occur more on metabolic enzymes than on other proteins. Additionally, metabolic enzymes (like other “housekeeping” proteins) tend to have slower turnover. The observation that acetylated peptidofoms tend to have slower turnover could simply be due to these two causally unrelated facts.

B) Acetylation might occur more frequently in the nucleus. Protein turnover might also be generally slower in the nucleus. These two causally unrelated effects could again explain the observations.

I am not saying that either A or B is true or even likely -- there are many additional possible interpretations. The essential point is that correlation (or “association” as they write above) does *not* imply any involvement.

Spurious correlations are a widespread phenomenon, as also illustrated here:

<https://www.tylervigen.com/spurious-correlations> I appreciate that the authors tried to use more careful wording in the revised manuscript. I also agree that many people in the field jump to premature conclusions, and the Wu et al. phospho turnover paper mentioned by the authors is a particularly bad example of this. However, I did not review the Wu et al paper, and the fact that other papers are worse in this regard does not make this manuscript sacrosanct.

2. Validation / follow up: The authors argue follow up experiments often require years of work including the creation of reagents, cellular models and specialized assays tailored to a single or few proteins. I fully agree, and this is exactly the type of work I expect to see in a high profile paper. I also agree with the authors that it is impossible to follow up in many different directions. However, instead of speculating about the function of many different modification sites in different proteins (as done in this manuscript), it would be more meaningful to speculate less and actually assess the function of one site. I do see the value in providing a resource for the community. However, without showing new function for at least one site, the functional relevance of the dataset remains unclear.

I suggested a number of follow-up experiments that would help characterize the function of

individual sites. While I do not expect the authors to follow any of these suggestions, I would have expected to see at least one informative follow-up experiment in the revised paper. The additional replicate they for FBWX7 is not the type of experiment I mean. I also find their explanation why the observed effect on TKT degradation is small plausible but again handwaving: For example, they say that they cannot distinguish between nuclear and cellular TKT in their whole cell lysate samples. I do not understand why they did not perform sub-cellular fractionation experiments to assess this in more detail. [1]
SEP

In summary, while I still think the dataset as such is of high quality, I think that publication in Nature communications also requires functional validation of the results obtained. I agree with the authors that this involves a lot of work, which is one reason why not every descriptive study ends up as a paper in a high profile journal. The authors argue in their point by point response that they compensate the lack of functional follow up by mentioning “rather many possible implications even if they are speculation at this stage.” This is exactly the problem I have with this paper: There is too much speculation and insufficient functional validation.

Point-to-point response to reviewer comments

Reviewer #1 (Remarks to the Author):

The authors have fully addressed all the points raised, and the manuscript can be recommended for publication.

Thank you for supporting publication of this work.

Reviewer #2 (Remarks to the Author):

The authors have done a good job of addressing the concerns highlighted by the reviewers, particularly the issues around correlation and causation.

While I was able to access the web portal (a great feature for interrogation of this dataset), it would be ideal if the authors were able to add a protein level search function in addition to the dropdown menu, allowing interrogation based on e.g. protein name.

Thank you for supporting publication of this work. The function requested is already available – you can just type the gene name or Swissprot identifier into the box after deleting the currently selected protein with the backspace key.

Reviewer #3 (Remarks to the Author):

I appreciate the detailed response by the authors. However, I do not think their revised manuscript adequately addresses the points I raised. I had three major points: the confusion of correlation and causation, the lack of validation and the fact that the method is not really new. The novelty of a method is less critical for a non-method journal, so I am only focussing on the other two points.

1. Correlation and causation: The authors still do not seem to see the fundamental flaw in the way they interpret their data. They say that they do not agree with my notion that the paper contains no data to support the view that Ubi and/or Ac can affect protein stability. In their point-by-point response they write: “While we may not provide formal proof regarding how this happens, the observation that e. g. acetylation is more often associated with slower than faster turnover implies an involvement of the PTM (regardless of cause or consequence).”

This is statement is wrong and highlights a fundamental misconception in how the authors interpret the data. Maybe this becomes clearer by these two alternative interpretations that come to my mind:

A) Acetylation could occur more on metabolic enzymes than on other proteins. Additionally, metabolic enzymes (like other “housekeeping” proteins) tend to have slower turnover. The observation that acetylated peptidofoms tend to have slower turnover could simply be due to these two causally unrelated facts.

The reviewer is indeed right that metabolic enzymes tend to be more stable when looking at the overall distribution of turnover values across the proteome. However, we account for this potential bias, particularly in figure 4, by normalizing the measured turnover of acetylated (and ubiquitinated) peptidofoms to their unmodified counterparts (peptides with the unmodified lysine residues; see turnover ratios/relative turnover in figure 4). Hence, we normalize the turnover of acetylated versions of these enzymes to the turnover of unmodified versions of the exact same enzymes precluding that the above stated fact impacts our analysis. In the main text associated with figure 4, we do not make any statements on acetylated or ubiquitinated forms for which we do not have information about the unmodified version.

B) Acetylation might occur more frequently in the nucleus. Protein turnover might also be generally slower in the nucleus. These two causally unrelated effects could again explain the observations.

This potential bias is also circumvented by the above stated strategy of normalizing the turnover of modified peptides to their unmodified counterpart peptides.

*I am not saying that either A or B is true or even likely -- there are many additional possible interpretations. The essential point is that correlation (or "association" as they write above) does **not** imply any involvement.*

Spurious correlations are a widespread phenomenon, as also illustrated here:

<https://www.tylervigen.com/spurious-correlations>; I appreciate that the authors tried to use more careful wording in the revised manuscript. I also agree that many people in the field jump to premature conclusions, and the Wu et al. phospho turnover paper mentioned by the authors is a particularly bad example of this. However, I did not review the Wu et al paper, and the fact that other papers are worse in this regard does not make this manuscript sacrosanct.

2. Validation / follow up: The authors argue follow up experiments often require years of work including the creation of reagents, cellular models and specialized assays tailored to a single or few proteins. I fully agree, and this is exactly the type of work I expect to see in a high profile paper. I also agree with the authors that it is impossible to follow up in many different directions. However, instead of speculating about the function of many different modification sites in different proteins (as done in this manuscript), it would be more meaningful to speculate less and actually assess the function of one site. I do see the value in providing a resource for the community. However, without showing new function for at least one site, the functional relevance of the dataset remains unclear. I suggested a number of follow-up experiments that would help characterize the function of individual sites. While I do not expect the authors to follow any of these suggestions, I would have expected to see at least one informative follow-up experiment in the revised paper. The additional replicate they for FBWX7 is not the type of experiment I mean. I also find their explanation why the observed effect on TKT degradation is small plausible but again handwaving: For example, they say that they cannot distinguish between nuclear and cellular TKT in their whole cell lysate samples. I do not understand why they did not perform sub-cellular fractionation experiments to assess this in more detail.

In summary, while I still think the dataset as such is of high quality, I think that publication in Nature communications also requires functional validation of the results obtained. I agree with the authors that

this involves a lot of work, which is one reason why not every descriptive study ends up as a paper in a high profile journal. The authors argue in their point by point response that they compensate the lack of functional follow up by mentioning “rather many possible implications even if they are speculation at this stage.” This is exactly the problem I have with this paper: There is too much speculation and insufficient functional validation.

Correlation and causation: We have scrutinized the manuscript for claims of causation and could not find any. Already in the abstract, we make a cautionary note: “While causal relationships may not always be immediately apparent, we hypothesize that PTMs with diverging turnover may distinguish states of differential protein stability, structure, localization, enzymatic activity, or protein-protein interactions.” The first half of that sentence implies that what we observe may not be causally related. In the second half, we merely state hypotheses. In our use of the word, a hypothesis is not a claim of causality but a statement of unproven validity which may be tested and, consequently, verified or falsified. We have gone over the text and used even more careful wording in places to make sure that readers understand that we offer possible/plausible explanations rather than facts. In addition, much of the discussion section is about being careful with the interpretation of PTM and protein turnover correlations and we make a very explicit statement in this regard: “Clearly, the molecular complexity of cellular systems makes the interpretation of turnover changes linked to protein modifications far from trivial. In particular, an observed faster or slower turnover of a PTM peptidofom does not imply that this difference is caused by the PTM.” We now added another sentence right behind: “In other words, beyond direct involvement of the PTMs, there are several alternative scenarios that might explain the observed correlation of certain PTMs with protein turnover”. We are, therefore, confident that readers are not misled to believe that correlations and causality are the same thing.

Lack of validation and novelty of the method: The results we report in this study are technically valid given the number replicates performed and basing all further data analysis on cases that pass the quality criteria provided in the main text and the methods section. The authors are of the opinion that adding functional validation experiments are beyond the scope of this already extensive manuscript. The current work goes very substantially beyond the state of the art and contains several novel aspects. First, it has not been experimentally demonstrated that the dynamic SILAC-TMT approach can be extended to the analysis of PTMs and showing that this is indeed the case is novel. Second, to the best of our knowledge, there is no published study yet that addresses ubiquitination and acetylation in the context of protein turnover. This is also new. There is only a single very recent report in the published literature (Wu et al. 2021, *Developmental Cell* 56) that investigated turnover and phosphorylation. We cited this work in our manuscript but note that it has a very different focus/interpretation compared to what we provide here. The revised manuscript points out more clearly how our study compares to Wu et al. Third, our manuscript is the first that comprehensively addresses three post-translational modifications (phospho, ubi, Ac) in the same study. Fourth, we also offer a novel interpretation of what differential turnover of PTMs may mean. While we do not claim that PTMs necessarily control a protein’s stability, we introduce the new concept that differential turnover of PTMs reflects differential states in which a protein can exist in a cell (e. g. localization, interaction, activity, co-occurrence with other functional PTMs etc.). A related but important new realization from the work is that PTMs with extremely different turnover cannot occur on the same protein molecule in a cell, re-enforcing the above notion that our approach can measure proteins in different (possibly functional) states. Fifth, we provide a website that contains all the PTM turnover data. This is a new and unique resource for the scientific community and building this online resource

represents a very substantial effort on the authors' part. It allows much more convenient interaction with the data than typically provided by static supplementary tables.